# Homogeneous Algorithms Can Reduce Competition in Personalized Pricing

**Nathanael Jo**
Massachusetts Institute of Technology
nathanjo@mit.edu

**Kathleen A. Creel**
Northeastern University
k.creel@northeastern.edu

**Ashia Wilson**
Massachusetts Institute of Technology
ashia@mit.edu

**Manish Raghavan**
Massachusetts Institute of Technology
mragh@mit.edu

## Abstract

Firms' algorithm development practices are often *homogeneous*. Whether firms train algorithms on similar data or rely on similar pre-trained models, the result is correlated predictions. In the context of personalized pricing, correlated algorithms can be viewed as a means to collude among competing firms, but whether or not this conduct is legal depends on the *mechanisms* of achieving collusion. We investigate the precise mechanisms through a formal game-theoretic model. Indeed, we find that *(1)* higher correlation diminishes consumer welfare and *(2)* as consumers become more price sensitive, firms are increasingly incentivized to compromise on the accuracy of their predictions in exchange for coordination. We demonstrate our theoretical results in a stylized empirical study where two firms compete using personalized pricing algorithms. Our results demonstrate a new mechanism for achieving collusion through correlation, which allows us to analyze its legal implications. Correlation through algorithms is a new frontier of anti-competitive behavior that is largely unconsidered by US antitrust law.

## 1 Introduction

Firms increasingly use algorithms for personalized pricing, charging some consumers more than others for the same good or service based on their perceived willingness to pay [59, 34, 4]. For example, a recent lawsuit against DoorDash alleged that the company charges Apple users more in delivery fees [51] and travel websites use browser information to charge US-based customers more [40]. Rather than targeting prices to individuals, a strategy called "first-degree price discrimination," firms segment consumers based on their characteristics and charge different prices to each segment, i.e., "third-degree price discrimination" [56, 48]. Today, firms use machine learning to segment consumers into more targeted categories [18]. For example, ridesharing competitors Uber and Lyft offer discounts to a subset of consumers based on algorithmic segmentation (see Figure 7).

Competing firms seek accurate personalized prices in order to better predict customers' willingness to pay. Offering a low price to a consumer willing to pay more means missing surplus value; offering a high price to a consumer unwilling to pay it means missing a sale altogether. However, if personalized prices are correlated between firms, firms are insulated from the competitive cost of their mistakes: if a firm failed to offer a discount, a correlated competitor is likely to also not offer a discount to that consumer [14]. In other words, firms can benefit from correlated prices.

39th Conference on Neural Information Processing Systems (NeurIPS 2025).

The rise of algorithmic pricing makes correlated predictions easier to achieve than ever.[1] Correlated predictions can occur when firms deploy algorithms made with "shared components" such as algorithms trained on similar datasets, aimed at similar benchmarks, or based on similar pre-trained models [8]. In other words, model development practices can be and indeed are fairly *homogeneous*. At the extreme, firms can adopt the same algorithm from a third party, creating an "algorithmic monoculture" with perfect predictive correlation across firms [31].

When firms benefit from correlated predictions, we might expect consumer welfare to be harmed. This naturally raises questions about how antitrust laws, which are designed to protect consumers from collusion, apply in a regime of algorithmic competition. Specifically, (when) does algorithmic correlation constitute illegal collusion on prices?

**Correlation and competition law.** An agreement between competitors to fix prices is *per se* illegal in the US, meaning that no further inquiry is needed as to the action's effect on the market or the parties' intent in reaching such an agreement [49, 54]. However, if multiple firms choose similar pricing models because they are the best models for business purposes – without intention to correlate – they are unlikely to be in violation of competition law. Actions such as changing prices in response to the market conditions are typically considered "parallel business behavior" and are not *per se* illegal without proof of intent to form an agreement or intent to reach supra-competitive prices [50, 33].

In the absence of explicit agreement to collude, courts often consider "plus factors" that can tip the scales from acceptable business behavior to illegal conduct. For example, evidence that firms act against their own economic self-interest is a common plus factor [32]. Critically, which pricing model would serve a firm's economic self-interest may depend on the model choices of other firms. A firm might profit from deploying a worse-performing model, apparently against its own self-interest, because it aligns with a competitor. Does this raise concerns about illegal conduct [29]? At this level of analysis, it remains unclear if or when adopting correlated algorithms constitutes illegal collusion—motivating the need for a formal model to capture the mechanisms at play.

**Our contributions.** We build a game-theoretic model of competition between two firms who use algorithms to price-discriminate. We find that, indeed, firms can benefit from correlating with competitors when consumers are sufficiently price sensitive (Theorem 4.2). We quantify this value as a trade-off between correlation and predictive performance.[2] Firms are sometimes willing to give up performance in exchange for correlation (Theorem 5.3), which is of concern because consumers are always worse off (more likely to pay higher prices) when competing firms' algorithms are more correlated (Theorem 4.1). We also conduct empirical analyses demonstrating that firms choosing different model classes or choosing to share training data may lead to correlated models in equilibrium.

Our findings allow us to develop insights about how current antitrust laws must evolve in the era of digital markets. In particular, the specter of frictionless algorithmic price correlation without overt communication should raise concerns about a new frontier of anti-competitive behavior. While the Federal Trade Commission (FTC) has brought multiple recent cases against parties using the same pricing algorithm to allegedly maintain inflated prices for hotel stays [52], rent [53], and pork [55], these cases involve traditional plus factors such as parties expressing their interest to collude in writing. Our work suggests the possibility of new plus factors such as broadcasting choice of model as an invitation to collude. We discuss these legal implications in Section 7.

In sum, we analyze algorithmic homogenization as a **novel mechanism for collusion**, which sets us apart from existing works on algorithmic collusion that often focus on reinforcement learning algorithms that can naturally find collusive equilibria [30, 42, 16, 23, 2]. Specifically, our results do not require assumptions about the particular learning dynamics of competing algorithms: even in the absence of threats of pricing retaliation or adaptive learning, the mere reduction of strategic uncertainty via correlated predictions is enough to drive higher prices in equilibrium.

---

[1]When we refer to "correlated algorithms", we mean algorithms that make correlated *errors*. Two fully independent algorithms with high accuracy will naturally be correlated with each other, but we are concerned with predictions that are even more correlated than the independent state of affairs.

[2]Though in practice firms may not directly face this trade-off, we compare correlation with performance because both are ways in which firms can extract more surplus from consumers. See Section 5.2 for a discussion.

## 2 Related Work

**Homogeneity and monoculture.** Our work builds on recent work in machine learning on "algorithmic monoculture", namely the state of affairs in which "many decision-makers rely on the same algorithm" and in doing so correlate their behavior [31]. Existing literature focuses on how monoculture harms the welfare of those who are subject to correlated algorithmic errors and face "homogeneous outcomes" [8, 28, 41]. Our work spotlights the harm to consumers that comes from higher prices in the context of personalized pricing.

**Economic models of oligopoly pricing.** We consider competition under a duopoly, which has been extensively studied in the economics literature. The works most related to ours are game-theoretic models of duopolies under Bayesian uncertainty [37, 12, 57, 26, 45, 1]. Much of this literature considers whether firms have incentives to collude by sharing information with one another. Whether a model will suggest that firms are rewarded for sharing information depends on a variety of modeling choices including whether firms compete over production quantity [15] or price [6]. In our model, as in these information-sharing models, firms' information is parameterized by its performance and degree of correlation. This allows us to reason about strategic decisions firms make regarding shared data, model components, or predictive algorithms.

**Personalized pricing.** A growing body of empirical, theoretical, and legal literature considers how personalized pricing interacts with concepts like competition and privacy [5, 19, 11, 21, 60, 13, 43]. Most related to our work are theoretical models of personalized pricing in the context of competition. Both Rhodes and Zhou [44] and Baik and Larson [3] consider models of competition in personalized pricing under first-degree price discrimination. In contrast, our model is designed to provide insights when firms have imperfect but potentially correlated information.

**Algorithmic collusion.** The spirit of modern antitrust law is to promote competition. There is generally broad consensus that an open and free market economy – which at its core fosters competition – benefits consumers by lowering prices, spurring innovation, and increasing the quality of goods and services [46, 38, 25, 9]. In the United States, antitrust enforcement relies on three sets of federal laws: the Sherman Act, the Clayton Act, and the FTC Act, each prohibiting different actions that harm competition. In this work, we will focus our attention to the parts of the Sherman Act and the FTC Act that are intended to delineate which forms of collusion among competitors are illegal.

In general, legal scholars consider three mechanisms for algorithmic collusion. First, an algorithm can be a tool that aids humans in explicitly sustaining cartel-like behavior. Second, an algorithm can be a hub that coordinates actions or be the sole algorithm used among competitors. Third, highly sophisticated algorithms can learn each other's behaviors and achieve supra-competitive prices without explicit communication. From the first to third category, the likelihood that the behavior is illegal decreases or, at best, the action becomes more likely to fall into a contested grey area [47]. This is because humans become less involved in achieving collusion, making it harder to prove an intent or conspiracy to agree to fix prices. Absent proof of intent to form an agreement, firms are considered to engage in **tacit collusion**, which is generally not illegal without plus factors.

Recent legal scholarship has raised concerns about the potential for algorithms to facilitate tacit collusion. Various works have proposed legal and legislative pathways to expand the powers of regulatory agencies [33] or methods to more effectively screen and audit for tacit collusion [36, 27]. However, the mechanisms for algorithmic tacit collusion have not been extensively studied. Several theoretical papers have found collusive outcomes under the third mechanism for algorithmic collusion, where reinforcement learning models compete in a repeated pricing setting [30, 42, 16, 23, 2]. In the repeated-game setting, it is well-known that collusive equilibria exist [24] since players can credibly threaten future retaliation. Past works have shown that standard learning paradigms can *find* such collusive equilibria. Our work, in contrast, studies the *existence* of collusive equilibria in the long-run given algorithms are ex ante correlated, meaning threats are not necessary to sustain them.

## 3 Model

We consider a duopoly model where two firms sell identical goods. For each consumer, a firm decides whether to offer a default price $H^r$ or a discounted price $L^r$.[3] Both firms incur the same unit costs $C$,

---

[3]This is consistent with pricing via "couponing" [e.g., 19], a strategy according to which firms target offers of fixed discounts (e.g., 20% off) to consumers.

Firm 2

|  | H | L |
|---|---|---|
| $[\tau_H]$ Firm 1   H | $(\frac{H}{2}, \frac{H}{2})$ | $(\sigma H, (1-\sigma)L)$ |
| L | $((1-\sigma)L, \sigma H)$ | $(\frac{L}{2}, \frac{L}{2})$ |

Firm 2

|  | H | L |
|---|---|---|
| $[\tau_L]$ Firm 1   H | $(0,0)$ | $(0, L)$ |
| L | $(L, 0)$ | $(\frac{L}{2}, \frac{L}{2})$ |

Table 1: Payoff matrices for both firms when the consumer is willing to pay the high price ($\tau_H$, top) and low price ($\tau_L$, bottom). Within each cell, we denote (Firm 1 payoff, Firm 2 payoff).

leading to a per-unit profit of $H = H^r - C$ and $L = L^r - C$ when pricing high and low, respectively. Without loss of generality, we assume that $C = 0$. We ignore consumers whose valuation $V$ for the good is less than $L$, and we define $\theta$ to be the fraction of consumers with valuation at least $H$.[4] We will use $\tau_H$ and $\tau_L$ to refer to consumers with valuations at least or strictly less than $H$ respectively.[5]

**Consumer behavior.** Consumers can purchase from either firm. Under perfect Bertrand competition, each consumer would simply choose to purchase the lower-priced good. The economics literature often relaxes the perfect competition assumption such that firms that price higher experience non-zero demand [see, e.g., 58]. This may be because firms have finite supply, meaning consumers are forced to purchase at a higher price when the low price goods sell out, or because some consumers are lazy and take the first price they encounter that is below their valuation. We parameterize this model as follows: When a consumer of type $\tau_H$ is offered a price $L^r$ by one firm and $H^r$ by the other, they purchase at price $H^r$ with probability $\sigma \in [0, 0.5]$ and $L^r$ with probability $1 - \sigma$. Thus, for larger values of $\sigma$, consumers are less price-sensitive.

We assume that consumers never pay a price above their valuation and always make a purchase as long as at least one firm offers a price below their valuation. Further, when a consumer is offered two identical prices, they choose a firm to purchase from uniformly at random. An intuitive example of this consumer behavior is riders choosing between ridesharing apps. When prices are the same, potential riders are indifferent between two rideshare services (i.e., they do not have brand loyalty). However, when prices differ and consumers are willing to pay the higher price, $\sigma$ models the friction consumers face in comparing the two options. Perhaps some consumers check both apps to shop for the lowest price, but others choose one app at random and take the first price below their valuation.

**Firms' utility and information structure.** Our consumer choice model yields the payoff matrices for the two firms for each consumer type shown in Table 1. Note that from firms' perspective, their utilities are with respect to unit profit as opposed to sale price. We will denote $U_i(\cdot; \tau)$ as the utility/payoff for firm $i$ for a given action profile and for a consumer's type $\tau \in \{\tau_L, \tau_H\}$. For example, $U_1((H, L); \tau_H) = \sigma H$ and $U_2((H, L); \tau_H) = (1-\sigma)L$.

Firms do not have perfect information and their algorithmic predictions make mistakes. We assume that when a consumer arrives with features $x$, each firm produces an algorithmic prediction $p_1(x), p_2(x) \in \{0, 1\}$, segmenting users into types $\{\tau_L, \tau_H\}$. For simplicity, we assume the algorithm has equal true positive and true negative rates, which we will denote $a_1$ for firm 1: $\mathbb{P}[p_1(x) = 1 \mid \tau_H] = \mathbb{P}[p_1(x) = 0 \mid \tau_L] = a_1$. We define the same quantity for firm 2 and will refer to $a$ as the model's performance. We will drop $x$ and simply refer to the algorithmic prediction as $p_1, p_2$.

An important feature of our model is that $p_1$ and $p_2$ need not be independent conditioned on user type. If, for example, both firms purchase data from a third party, their predictions may be correlated. Throughout the paper, the terms "correlated predictions" and "correlated algorithms" strictly mean that algorithms make *correlated errors*. Two fully independent algorithms with high accuracy will naturally be correlated with each other, but we are concerned with algorithms that become even more correlated than the independent state of affairs. In the extreme case of algorithmic monoculture, firms use the same model, i.e.., their predictions are identical regardless of accuracy. We parameterize their correlation by $\rho \in [0, 1]$, where $\rho = 0$ implies independence ($p_1 \perp\!\!\!\perp p_2 \mid \tau$) and $\rho = 1$ implies maximal correlation.[6] When $a_1 = a_2$, $\rho = 1$ if and only if $p_1 = p_2$ deterministically. For now, we

---

[4]We treat $H$ and $L$ as exogenous to this model. We interpret $H$ as a posted price (e.g., the nominal fare offered by a ride-sharing app) and $L$ as a fixed discount (e.g., a coupon) on that posted price.

[5]While a more sophisticated model might directly estimate consumer willingness to pay, firms in practice simplify continuous prediction problems into discrete ones [39] and collect data at discrete price points [18].

[6]When $a_1 \neq a_2$, $p_1$ and $p_2$ cannot be perfectly correlated. See Appendix A for a formal definition of $\rho$.

treat $\rho$ as exogenous; we will consider strategic choices impacting $\rho$ in Section 5. We assume all parameters are known to both firms.[7] In total, our model has five free parameters (see Table 2).

**Equilibrium concept.** A firm's strategy space is simple: for each binary prediction ($p \in \{0, 1\}$) given by the algorithm, set a price $\{L, H\}$. This results in 4 possible (pure) strategies. Because all parameters are known, firms know the joint distribution on $p_1, p_2, \tau$. Our analysis will focus on Bayes Nash Equilibria (BNE). We do not require that firms price based on the algorithm's predictions; for some parts of the parameter space, firms may ignore the algorithm and either always or never offer the discount. We will focus on the region where both firms follow their algorithms at equilibrium (i.e., price-discriminate), formally: $s^*(p) = H$ if $p = 1$, $L$ if $p = 0$.

The strategy profile $(s^*, s^*)$ (both firms price-discriminate) is an equilibrium if and only if the conditions below hold:

$$\mathbb{E}_{p_2,\tau} \left[ U_1((H, s^*(p_2)); \tau) \mid p_1 = 1 \right] \geq \mathbb{E}_{p_2,\tau} \left[ U_1((L, s^*(p_2)); \tau) \mid p_1 = 1 \right]$$

$$\mathbb{E}_{p_2,\tau} \left[ U_1((L, s^*(p_2)); \tau) \mid p_1 = 0 \right] \geq \mathbb{E}_{p_2,\tau} \left[ U_1((H, s^*(p_2)); \tau) \mid p_1 = 0 \right].$$

Analogous conditions must hold for player 2. Intuitively, expected utility when both firms follow the algorithm's recommendation (both when $p_1 = 1$ and when $p_1 = 0$) must be higher than when one firm deviates. We are only interested in the conditions where $(s^*, s^*)$ is a BNE because firms should follow the algorithm if they adopt it in the first place. In Ap-

| Parameter | Interpretation |
|---|---|
| $\theta \in [0, 1]$ | Frac. of consumers w/ demand $\geq H^r$ |
| $a_1, a_2 \in [0.5, 1]$ | Model performance for firms 1 & 2 |
| $\sigma \in [0, 0.5]$ | Consumers' indifference to price |
| $\rho \in [0, 1]$ | Degree of model correlation or homogenization |

Table 2: List of free parameters in the model.

pendices D and E, we expand on our model by investigating settings where consumers have brand loyalty to a firm and where $n$ firms compete in a market.

Importantly, we note that BNE captures the long-run behavior of firms' algorithmic choices rather than the outcome of a single static pricing round. We never specify *how* firms might converge to that equilibrium, unlike previous works [30, 42, 16, 23, 2] that model reinforcement learning or repeated-game dynamics. Collusive equilibria always exist in repeated games (as folk theorems suggest [24]), but do not necessarily exist in our setting. Further, while these works specify conditions for convergence, our equilibrium concept uniquely allows us to isolate the precise mechanism that yields supra-competitive prices: the degree of correlation between competing algorithms.

## 4 Main Results

We find that *(1)* consumers are worse off as algorithms become more correlated; and *(2)* firms exhibit stronger preferences for correlation as consumers are price sensitive.

**(1) Consumers are always worse off when pricing algorithms are correlated.** When firms' pricing strategies are correlated (e.g., they price identically), consumers have less choice and must accept the given price or forgo the good. Conversely, when firms price independently, consumers are more likely to have the option to choose a lower price. We formalize this in Theorem 4.1.

**Theorem 4.1.** *Fix $\sigma$, $a_1$, $a_2$, $\theta$, and $H/L$. For all $\rho$ such that $(s^*, s^*)$ is a BNE, consumer welfare is decreasing in $\rho$.*

All proofs can be found in Appendix B. Our next few results describe when firms benefit by choosing correlated algorithms (and thereby harming consumers).

**(2) Higher consumer price sensitivity leads to a stronger firm preference for correlation.** As consumers become more price sensitive ($\sigma$ decreases), firms increasingly prefer to use more correlated algorithms over independent ones.

**Theorem 4.2.** *Suppose, for fixed $\theta, a_1, a_2$, and $R = H/L$, $(s^*, s^*)$ is a BNE when $\rho = \rho_A$ and $\rho = \rho_B$, with $\rho_B > \rho_A$. Assuming both $a_1, a_2 < 1$, firms have higher utility under $\rho_B$ when $\sigma < \sigma^*(\theta, R)$, where $\sigma^*(\theta, R) = \frac{R\theta - 1}{2\theta(R-1)}$. Otherwise, firms have weakly higher utility under $\rho_A$.*

---

[7]This assumption is especially common in oligopolies with few players that interact with each other frequently.

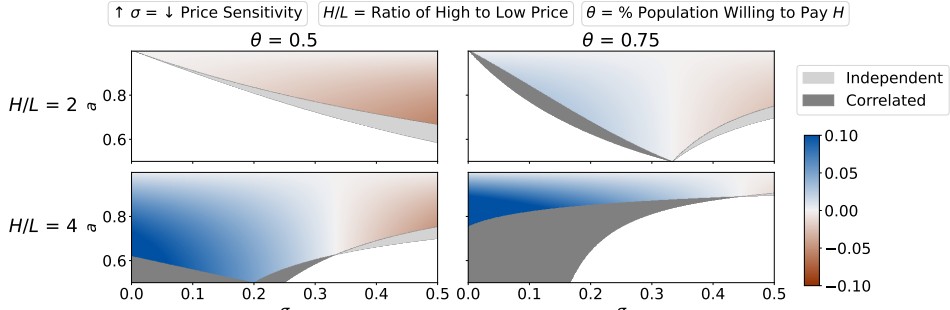

Figure 1: Regions where firms following the algorithm's recommendation is a Bayes Nash Equilibrium (BNE) for independent models only ($\rho = 0$, light gray), identical models only ($\rho = 1$, dark gray), and both (gradient). The gradient represents the difference in firm utility when $\rho = 1$ relative to $\rho = 0$; blue (red) signifies positive (negative) difference. Columns represent two values of $\theta \in \{0.5, 0.75\}$, while rows represent two values of $H/L \in \{2, 4\}$. The x-axis in each subfigure is $\sigma$ and the y-axis is $a = a_1 = a_2$.

Intuitively, when consumers are more price sensitive, firms have a higher risk in pricing $H$ because they may get undercut by their competitor and only attain a minority of the market. In these situations, firms *prefer correlation* because there is no risk of undercutting; both firms receive similar prediction and therefore price similarly. On the other hand, conditioned on pricing low, firms *prefer independence*: a firm would rather be undercutting its competitor than pricing identically. The balance between these two competing forces—a preference for correlation when pricing high and a preference for independence when pricing low—determine whether a firm prefers correlation overall.

The tension between these forces is mediated by $\sigma$, which determines the relative risk from being undercut. Indeed, in Figure 1 we observe that within the gradient region (where both independent and correlated models are equilibria), preference for correlation monotonically decreases (from blue to red) as $\sigma$ increases. In the extreme case when $\sigma = 0.5$ (consumers are completely price insensitive), firms always prefer independence. When a firm predicts $p_i = 1$ and prices $H$ accordingly, there is zero risk in being undercut: the firm receives $0.5H$ if $\tau = \tau_H$ and $0$ otherwise, regardless of their opponent's price. However, when a firm predicts $p_i = 0$ and prices $L$, they would in fact prefer that their opponent prices $H$ so that they guarantee a sale when the consumer's valuation is low ($\tau = \tau_L$).

## 5   Strategic Choices in Algorithm Development

We have shown that correlation can raise firm utility. Next, we examine its impact on strategic decisions in algorithm development *before* price competition, illustrating one possible strategic choice among many.

### 5.1   Model

Two firms choose between two model development processes: *(1)* collecting their own training data, yielding a model $p_i$ with performance $a_i$, or *(2)* purchasing training data from the same vendor, producing a model $p_c$ with performance $a_c$. When both firms buy data, their models are correlated at $\rho = \rho_c > 0$. If a firm collects its own data, we assume their model's errors are independent of their competitor's ($\rho = \rho_0 = 0$), though in practice, independent data may not guarantee uncorrelated errors. More broadly, any shared component in model development—not just data procurement—can induce correlation. For example, our experiments in Section 6 allow firms to choose between model classes with varying levels of correlation. To summarize, firms have the following payoff matrix:

|  |  | Firm 2 | |
|---|---|---|---|
|  |  | $p_c$ | $p_i$ |
| Firm 1 | $p_c$ | $E_{\rho_c}(p_c, p_c)$ | $E_{\rho_0}(p_c, p_i)$ |
|  | $p_i$ | $E_{\rho_0}(p_i, p_c)$ | $E_{\rho_0}(p_i, p_i)$ |

where, for instance, $E_{\rho_c}(p_c, p_c) =$
$$\Big( \mathbb{E}_{p_c, p_c, \tau; \rho=\rho_c} \left[ U_1((s^*(p_c), s^*(p_c)); \tau) \right],$$
$$\mathbb{E}_{p_c, p_c, \tau; \rho=\rho_c} \left[ U_2((s^*(p_c), s^*(p_c)); \tau) \right] \Big)$$

We are interested in analyzing conditions under which equilibria exist. Two possible equilibria are *(1)* both firms choose algorithm $p_c$ with correlation $\rho_c$, and *(2)* both firms choose algorithm $p_i$ with

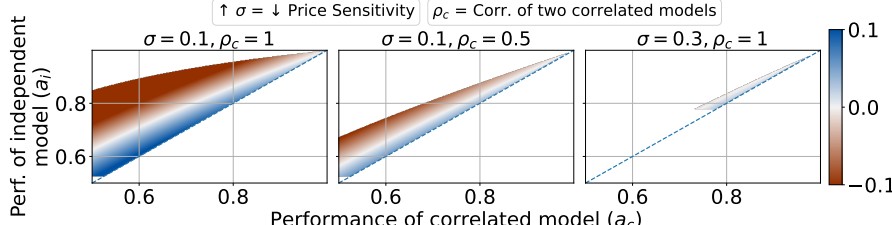

Figure 2: Regions where firms using both correlated models and independent models are Pure Nash Equilibra (first-stage game). An additional condition is that firms following the algorithm's recommendation must be a Bayes Nash Equilibrium (second-stage game). The x-axis is the performance of the correlated algorithm $a_c$, and the y-axis is the performance of the independent algorithm $a_i$. The gradient represents the difference in firm utility when $\rho = \rho_c$ (correlated) at performance $a_c$ relative to the utility at $\rho = 0$ (independent) at performance $a_i$; blue (red) signifies positive (negative) difference. All subfigures show parameters for which firms have a preference for correlation at $a_c = a_i$ as per Theorem 4.1, with $H/L = 3, \theta = 0.75$.

independent outcomes $\rho = \rho_0 = 0$. From hereon, **we will refer to scenario (1) and (2) respectively as "correlated" and "independent"**, ignoring that other actions can also lead to independence.

Formally, the following conditions must hold for correlation or independence to be Pure Nash Equilibria (PNE):

correlated in equilibrium:   $E^1_{\rho_c}(p_c, p_c) \geq E^1_{\rho_0}(p_i, p_c)$ and $E^2_{\rho_c}(p_c, p_c) \geq E^2_{\rho_0}(p_c, p_i)$

independent in equilibrium:   $E^1_{\rho_0}(p_i, p_i) \geq E^1_{\rho_0}(p_c, p_i)$ and $E^2_{\rho_0}(p_i, p_i) \geq E^2_{\rho_0}(p_i, p_c)$.

As in the previous section, we focus on the strategy $s^*$ of price-discriminating. As such, an additional condition for equilibrium is that $(s^*, s^*)$ is a BNE in the downstream second-stage game.

## 5.2   Results

Our main result shows that under certain conditions, firms may prefer correlation over independence, even when the correlated algorithm performs worse. Crucially, the following theorems apply only in parameter spaces where firms adopt the price-discriminating strategy $s^*$.

**Lemma 5.1.** *When $a_i > a_c$, both firms choosing independence is always a PNE.*

We next establish the conditions under which both firms correlating are in equilibrium, which comes from Theorem 4.2.

**Corollary 5.2** (Corollary to Thm 4.2). *For $a_i = a_c$ and $\sigma < \sigma^*(\theta, R)$, correlating is strictly a PNE.*

With Lemma 5.1 and Corollary 5.2 in hand, we can now state our final result.

**Theorem 5.3.** *For $\sigma < \sigma^*(\theta, R)$ and any $a_c$, there exists $a_i$ such that both correlation and independence are PNE **and** firms have higher utility under correlation than under independence.*

Theorem 5.3 says that given a preference for correlation at $a_i = a_c$, there are settings where firms derive strictly higher utility from an equilibrium with correlated but less accurate models. We will demonstrate this effect in Section 6.

Figure 2 shows the various regions where both correlation and independence are PNE. All subfigures depict model parameters where firms prefer correlation at $a_i = a_c$. As expected, all subfigures have a region at $a_i = a_c + \epsilon, \epsilon > 0$ where correlation is still preferred to independence despite having a lower performance (blue gradient region). It seems that higher price sensitivity and a higher correlation option tend to increase the valid region of $\epsilon$. For example, when $\theta = 0.75, H/L = 3, \sigma = 0.1$, and $\rho_c = 1$, firms would rather correlate at a performance of $a_c = 0.6$ than have a much more informative independent model of $a_i = 0.72$.

**Trade-off between correlation and performance.** While firms may not have direct information about a model's correlation with competitors, development practices can still favor correlation. Beyond simply measuring predictive performance, firms typically **A/B test** new models when deploying them. Having developed a seemingly more accurate model, a firm might run an A/B test that reveals that it leads to *lower* profits, since it happens to correlate less with a competitor. Thus, market signals suffice to enable firms to trade off accuracy for correlation.

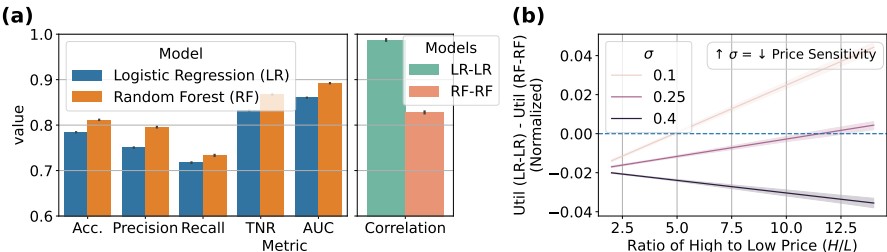

Figure 3: **(a)** [Left] Accuracy, precision, recall, true negative rate (TNR), and area under ROC curve (AUC) for a given firm deploying a logistic regression (LR) or random forest (RF) model. Since both firms 1 and 2 face the same model options, their results are identically distributed. [Right] Correlation between both firms' models when they both use logistic regression (LR-LR) or both use random forests (RF-RF). Error bars indicate 95% confidence intervals over 15 seeds. **(b)** Utility when both firms use logistic regression models (LR-LR) subtracted by utility when both firms use random forests (RF-RF). Greater than 0 indicates a preference for correlation at the expense of predictive performance. $x$-axis varies the proportion of $H$ (high price) to $L$ (low price), and line colors indicate different values of $\sigma$, where a lighter color means higher consumer sensitivity to price. Shaded region indicates 95% confidence intervals over 15 seeds.

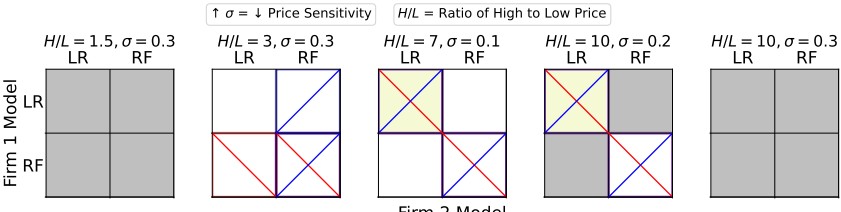

Figure 4: Best response matrices for the two firms where the action space is to deploy a logistic regression (LR) or random forest (RF) model, over five model parameters. Best response for Firms 1 and 2 are highlighted in blue and red. Nash equilibria exist when both blue and red are highlighted in the same box (e.g., (LR, LR) in the middle subfigure). When both (LR, LR) and (RF, RF) are equilibria, a yellow square indicates higher firm utility between the two. Grey boxes are "invalid" regions because $(s^*, s^*)$ would not have been a BNE in the downstream game where firms compete on prices. These results use the average firm utility over 15 seeds.

## 6 Empirical Study

We demonstrate our theoretical results in a stylized game between two firms that predict income based on demographic attributes. We use ACSIncome data [17], which contains US Census data from 2018. The task is to predict whether an individual earns over $50,000, which aligns with pricing via "couponing" [19].

### 6.1 Setup: Different Model Classes

To further illustrate strategic choices in our model, we consider how firms select different model classes. In Appendix C.2, we conduct experiments varying the degree to which firms train their models on shared data. Our experiment involves two firms. Each firm chooses between a better performing (random forests) and worse performing (logistic regression) model. However, logistic regression – despite having worse performance – has lower variance, meaning that it is likely to be more correlated when the opposing firm also chooses the same model class. We will show empirically that firms may prefer to sacrifice predictive performance in exchange for correlation.

Both firms train and test on Census data in California. The test set is 30% of the data ($n = 58,700$) and is fixed across both firms. We randomly split half of the remaining 70% as the training set for Firm 1, and the other half for Firm 2, each having 35% of the entire data to train ($n = 68,482$). We repeat the training data splits over 15 random seeds.

Figure 3(a) shows the performance and correlation for both firms when using a logistic regression (LR) and a random forest (RF) model. RF outperforms LR across many metrics: accuracy, precision, recall, true negative rate, and area under ROC curve. This is by design – our goal is to simulate a

scenario where firms have a choice between a more correlated model with worse performance (LR) and a better performing but less correlated model (RF). See Appendix C.1 for details.

## 6.2 Results

**Preference for correlation.** Figure 4 shows best response matrices for both firms choosing between LR or RF, over various values of $\sigma$ and $H/L$. Cells with a blue and red cross indicate a Pure Nash Equilibrium (PNE) for that action profile. When $H/L$ is too low or too high, firms will never choose to follow their algorithms to begin with (grey cells) because always pricing low or high will give a higher expected utility. When $H/L$ is moderate, less correlation (RF, RF) is always a PNE as per Theorem 5.1. More correlation (LR, LR) is a PNE under the condition outlined in Corollary 5.2. Finally, when both (LR, LR) and (RF, RF) are PNE, the difference in performance between LR and RF are small enough such that (LR, LR) is higher in utility (yellow cell) than (RF, RF) as per Theorem 5.3. This preference for correlation (LR, LR) occurs when $H/L$ is large and $\sigma$ is low (consumers are more price sensitive), as Figure 3(b) illustrates. As per Theorem 4.2, correlation is most beneficial to firms when there is a high risk of being undercut by the opponent; therefore, firms would rather have certainty about the other firm's actions than a better performing model.

**Firms prefer lower variance models under competition.** Lower variance models have less predictive multiplicity [7], and thus predictive errors are more correlated. Our empirical study suggests that competing firms are pushed to adopt simpler (higher bias, lower variance) models on the margins.

# 7 Discussion

Taken together, our results suggest that firms will sometimes prefer a less accurate personalized pricing algorithm when doing so allows them to better correlate their behavior with their competitors (Theorem 5.3) and this behavior reduces consumer welfare (Theorem 4.1). Furthermore, firms are more likely to prefer correlated algorithms when consumers are price sensitive (Theorem 4.2) and the consumers most likely to suffer are those to whom the price matters most.

**Correlation is a mechanism to reduce competition and sustain higher prices.** When firms make up a duopoly, using more correlated algorithms allows firms to reduce competition, which increases prices. When algorithms are not correlated, firms naturally attempt to undercut their opponent in order to extract more surplus, so the high price equilibrium cannot be sustained. This undercutting will continue to lower prices until firms reach a new equilibrium.

**Models can become correlated when any part of the development pipeline is homogeneous**, such as using similar pre-trained models, lower variance models (Section 6), or training on similar data (Appendix C.2). We empirically demonstrate that these settings lead to higher prices.

We note several limitations. Our findings and discussions on antitrust law apply specifically to algorithmic pricing. Moreover, our stylized model is intended to illustrate the possibility of correlated outcomes occurring under reasonable conditions. Future work may attempt to investigate more mechanisms or even empirically investigate this possibility in real-life markets.

## 7.1 Legal Implications

Our results add to the growing body of work suggesting that the ease of collusion that algorithmic price-setting facilitates may support a revision of traditional anti-trust standards [35, 22, 33].

**Publicly signaling models may invite collusion.** Mazumdar [33] suggests that adopting a pricing algorithm that "broadcasts" its intentions can signal an invitation to collude. While there is little precedent for firms publicly committing to a model, our findings suggest this may pose a risk.

In particular, our model suggests that publicly adopting a *less accurate* model could be considered an invitation to collude. Assume that both correlated models (LR, LR) and independent models (RF, RF) are equilibria (e.g., middle subfigure of Figure 2), and firms initially use (RF, RF). In order for firms to reach the collusive outcome of (LR, LR) without explicit communication or agreement, one firm must unilaterally switch to LR, sacrificing its own utility (by leaving an equilibrium) in the hope that its competitor will follow. This costly action functions as a signal—demonstrating a willingness to reduce competition despite short-term losses. Antitrust law should determine whether

public announcement of model choice can be an anti-competitive "plus factor" in the same way that public announcement of intent to price high can be anti-competitive.

Choosing a less accurate model is not the only way to collude by correlating. Among *equally accurate* models, a firm selecting the model most correlated with a competitor would not necessarily constitute a "plus factor" since choosing a best-in-class model aligns with a firm's economic self-interest. As discussed, however, *intent* to achieve supracompetitive prices is sufficient to establish illegal collusion. Thus, choosing a correlated model may be concerning if firms intended to reduce competition.

**Intentional choice of correlated models as a frontier of competition law.** Intention to collude on prices has long been a cornerstone of anti-trust law. We propose that intentionally adopting correlated algorithms can constitute illegal collusion, just as intent to coordinate on higher prices does. Previous legal cases alleging algorithmic collusion like RealPage [53] and AgriStats [55] have relied on a hub-and-spoke structure, where firms share information centrally, receive price recommendations, and face enforcement for deviations. Recently, the state of California updated its antitrust laws to prohibit the use of a common pricing algorithm that intends to restrain trade [10]. However, our model demonstrates that horizontal collusion can occur even without a central coordinator — firms merely need to knowingly homogenize.

## Acknowledgments and Disclosure of Funding

This research was supported in part by an MIT Social and Ethical Responsibilities of Computing (SERC) seed grant.

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

## A  Model (Continued)

### A.1  Correlation Parameter

We parameterize correlation between two models $p_1$ and $p_2$ with $\rho \in [0, 1]$, see Table 3 for the joint distribution on $\mathbb{P}[p_1, p_2, \tau]$. Note that in the case where $\rho = 1$ and $a_1 = a_2$, we are modeling a scenario where both firms are using the same algorithm (i.e., monoculture).

| $\tau$ | $p_1$ | $p_2$ | $\mathbb{P}[p_1, p_2, \tau]$ |
|---|---|---|---|
| $\tau_H$ | 1 | 1 | $\theta[a_1 a_2 + \rho(\min(a_1, a_2) - a_1 a_2)]$ |
| $\tau_H$ | 1 | 0 | $\theta[a_1(1 - a_2) - \rho(\min(a_1, a_2) - a_1 a_2)]$ |
| $\tau_H$ | 0 | 1 | $\theta[(1 - a_1)a_2 - \rho(\min(a_1, a_2) - a_1 a_2)]$ |
| $\tau_H$ | 0 | 0 | $\theta[1 - a_1 - a_2 + a_1 a_2 + \rho(\min(a_1, a_2) - a_1 a_2)]$ |
| $\tau_L$ | 1 | 1 | $(1 - \theta)[1 - a_1 - a_2 + a_1 a_2 + \rho(\min(a_1, a_2) - a_1 a_2)]$ |
| $\tau_L$ | 1 | 0 | $(1 - \theta)[(1 - a_1)a_2 - \rho(\min(a_1, a_2) - a_1 a_2)]$ |
| $\tau_L$ | 0 | 1 | $(1 - \theta)[a_1(1 - a_2) - \rho(\min(a_1, a_2) - a_1 a_2)]$ |
| $\tau_L$ | 0 | 0 | $(1 - \theta)[a_1 a_2 + \rho(\min(a_1, a_2) - a_1 a_2)]$ |

Table 3: Joint distribution $\mathbb{P}[p_1, p_2, \tau]$.

## B  Proofs

### B.1  Consumer Welfare and Proof of Theorem 4.1

Before proving the theorem, we will introduce some additional notation. Let $W((\cdot); \tau)$ denote consumer welfare under the action profile $(\cdot)$ and demand state $\tau$. We define consumer welfare as the consumer valuation of the good subtracted by the cost of the good. As such, we define two additional variables $V_L, V_H$ to be consumers' expected valuation under $\tau_L$ and $\tau_H$, respectively. Let $\delta_L = V_L - L^r$ and similarly for $\delta_H = V_H - H^r$. By definition, $\delta_L, \delta_H \geq 0$ – otherwise, consumers will not purchase the good. Consumer welfare under the various actions and demand states can be summarized in Table 4.

|  |  | $\tau_H$ Firm 2 | | $\tau_L$ Firm 2 | |
|---|---|---|---|---|---|
|  |  | $H$ | $L$ | $H$ | $L$ |
| Firm 1 | $H$ | $\delta_H$ | $\delta_H + (1 - \sigma)(H^r - L^r)$ | $0$ | $\delta_L$ |
|  | $L$ | $\delta_H + (1 - \sigma)(H^r - L^r)$ | $\delta_H + (H^r - L^r)$ | $\delta_L$ | $\delta_L$ |

Table 4: Consumer welfare under all action possibilities and both demand states $(\tau_H, \tau_L)$.

*Proof.* We will denote expected consumer welfare for a given value of $\rho$ as

$$\mathbb{E}_{p_1, p_2, \tau; \rho}[W((s^*(p_1), s^*(p_2)); \tau)].$$

Our goal is to show that

$$\frac{d}{d\rho} \mathbb{E}_{p_1, p_2, \tau; \rho}[W((s^*(p_1), s^*(p_2)); \tau)] < 0.$$

Our approach will be to show that increasing $\rho$ increases the likelihood that $p_1 = p_2$, which in turn reduces consumer welfare. First, observe that

$$\mathbb{E}_{p_1, p_2, \tau; \rho}[W((s^*(p_1), s^*(p_2)); \tau)] = \mathbb{E}_{p_1, p_2, \tau; \rho}[W((s^*(p_1), s^*(p_2)); \tau) \mid p_1 = p_2] \Pr_{p_1, p_2, \tau; \rho}[p_1 = p_2]$$

$$+ \mathbb{E}_{p_1, p_2, \tau; \rho}[W((s^*(p_1), s^*(p_2)); \tau) \mid p_1 \neq p_2] \Pr_{p_1, p_2, \tau; \rho}[p_1 \neq p_2].$$

We will show that

$$\mathbb{E}_{p_1, p_2, \tau; \rho}[W((s^*(p_1), s^*(p_2)); \tau) \mid p_1 = p_2]$$

and

$$\mathbb{E}_{p_1,p_2,\tau;\rho}\left[W((s^*(p_1), s^*(p_2)); \tau) \mid p_1 \neq p_2\right]$$

do not depend on $\rho$.

$$\begin{aligned}
\mathbb{E}_{p_1,p_2,\tau;\rho}\left[W((s^*(p_1), s^*(p_2)); \tau) \mid p_1 = p_2\right] &= \mathbb{E}_{p_1,p_2,\tau;\rho}\left[W((s^*(p_1), s^*(p_2)); \tau) \mid p_1 = p_2, \tau = \tau_H\right] \Pr_{p_1,p_2,\tau;\rho}\left[\tau_H \mid p_1 = p_2\right] \\
&\quad + \mathbb{E}_{p_1,p_2,\tau;\rho}\left[W((s^*(p_1), s^*(p_2)); \tau) \mid p_1 = p_2, \tau = \tau_L\right] \Pr_{p_1,p_2,\tau;\rho}\left[\tau_L \mid p_1 = p_2\right] \\
&= W((s^*(p_1), s^*(p_2)); \tau_H) \Pr_{p_1,p_2,\tau;\rho}\left[\tau_H \mid p_1 = p_2\right] \\
&\quad + W((s^*(p_1), s^*(p_2)); \tau_L) \Pr_{p_1,p_2,\tau;\rho}\left[\tau_L \mid p_1 = p_2\right]
\end{aligned}$$

Note that by definition, $\Pr_{p_1,p_2;\rho}[p_1 = p_2 \mid \tau = \tau_H] = \Pr_{p_1,p_2;\rho}[p_1 = p_2 \mid \tau = \tau_L] = \Pr_{p_1,p_2,\tau;\rho}[p_1 = p_2]$. Therefore,

$$\Pr_{p_1,p_2,\tau;\rho}[\tau = \tau_L \mid p_1 = p_2] = \frac{\Pr_{\tau;\rho}[\tau = \tau_L] \Pr_{p_1,p_2;\rho}[p_1 = p_2 \mid \tau = \tau_L]}{\Pr_{p_1,p_2;\rho}[p_1 = p_2]} = 1 - \theta$$

$$\Pr_{p_1,p_2,\tau;\rho}[\tau = \tau_H \mid p_1 = p_2] = \frac{\Pr_{\tau;\rho}[\tau = \tau_H] \Pr_{p_1,p_2;\rho}[p_1 = p_2 \mid \tau = \tau_H]}{\Pr_{p_1,p_2;\rho}[p_1 = p_2]} = \theta$$

This implies

$$\mathbb{E}_{p_1,p_2,\tau;\rho}\left[W((s^*(p_1), s^*(p_2)); \tau) \mid p_1 = p_2\right] = \theta W((s^*(p_1), s^*(p_2)); \tau_H) + (1 - \theta)W((s^*(p_1), s^*(p_2)); \tau_L).$$

A similar argument shows that $\mathbb{E}_{p_1,p_2,\tau;\rho}[W((s^*(p_1), s^*(p_2)); \tau) \mid p_1 \neq p_2]$ does not depend on $\rho$.
Next, we will show that that

$$\mathbb{E}_{p_1,p_2,\tau}\left[W((s^*(p_1), s^*(p_2)); \tau) \mid p_1 = p_2\right] \leq \mathbb{E}_{p_1,p_2,\tau}\left[W((s^*(p_1), s^*(p_2)); \tau) \mid p_1 \neq p_2\right], \quad (1)$$

meaning that consumers have higher expected utility when offered different prices. Because $\tau$ is independent of the event $p_1 = p_2$, we can analyze each $\tau \in \{\tau_L, \tau_H\}$ separately. For $\tau_H$,

$$\begin{aligned}
\mathbb{E}_{p_1,p_2}\left[W((s^*(p_1), s^*(p_2)); \tau_H) \mid p_1 = p_2, \tau = \tau_H\right] - \delta_H &= \Pr_{p_1,p_2}[p_1 = p_2 = 1 \mid p_1 = p_2, \tau = \tau_H](W((H,H); \tau_H) - \delta_H) \\
&\quad + \Pr_{p_1,p_2}[p_1 = p_2 = 0 \mid p_1 = p_2, \tau = \tau_H](W((L,L); \tau_H) - \delta_H) \\
&= \Pr_{p_1,p_2}[p_1 = p_2 = 1 \mid p_1 = p_2, \tau = \tau_H] \cdot 0 \\
&\quad + \Pr_{p_1,p_2}[p_1 = p_2 = 0 \mid p_1 = p_2, \tau = \tau_H](H^r - L^r) \\
&= \frac{1 - a_1 - a_2 + (1-\rho)a_1 a_2 + \rho \min(a_1, a_2)}{1 - a_1 - a_2 + 2(1-\rho)a_1 a_2 + 2\rho \min(a_1, a_2)}(H^r - L^r) \\
&\leq \frac{1}{2}(H^r - L^r)
\end{aligned}$$

because $a_1$ and $a_2$ are both at least 0.5. Similarly,

$$\mathbb{E}_{p_1,p_2}\left[W((s^*(p_1), s^*(p_2)); \tau_H) \mid p_1 \neq p_2, \tau = \tau_H\right] - \delta_H = (1 - \sigma)(H^r - L^r)$$

$$\geq \frac{1}{2}(H^r - L^r)$$

since $\sigma \leq 0.5$, meaning

$$\mathbb{E}_{p_1,p_2}\left[W((s^*(p_1), s^*(p_2)); \tau_H) \mid p_1 = p_2, \tau = \tau_H\right] \leq \mathbb{E}_{p_1,p_2}\left[W((s^*(p_1), s^*(p_2)); \tau_H) \mid p_1 \neq p_2, \tau = \tau_H\right],$$

$$(2)$$

Next, observe that

$$\mathbb{E}_{p_1,p_2}\left[W((s^*(p_1), s^*(p_2)); \tau_L) \mid p_1 = p_2, \tau = \tau_L\right] \leq \mathbb{E}_{p_1,p_2}\left[W((s^*(p_1), s^*(p_2)); \tau_L) \mid p_1 \neq p_2, \tau = \tau_L\right]$$

$$(3)$$

simply because the left hand side is at most $\delta_L$ and the right hand side is deterministically $\delta_L$. Combining (2) and (3) and using the fact that $\tau$ is independent of the event $p_1 = p_2$ proves (1). As a result,

$$
\frac{d}{d\rho} \mathop{\mathbb{E}}_{p_1,p_2,\tau;\rho} [W((s^*(p_1), s^*(p_2)); \tau)] = \frac{d}{d\rho} \left( \mathop{\mathbb{E}}_{p_1,p_2,\tau} [W((s^*(p_1), s^*(p_2)); \tau) \mid p_1 = p_2] \mathop{\Pr}_{p_1,p_2,\tau;\rho} [p_1 = p_2] \right.
$$

$$
+ \mathop{\mathbb{E}}_{p_1,p_2,\tau} [W((s^*(p_1), s^*(p_2)); \tau) \mid p_1 \neq p_2] \left( 1 - \mathop{\Pr}_{p_1,p_2,\tau;\rho} [p_1 = p_2] \right) \Bigg)
$$

$$
= \left( \mathop{\mathbb{E}}_{p_1,p_2,\tau} [W((s^*(p_1), s^*(p_2)); \tau) \mid p_1 = p_2] \right.
$$

$$
\left. - \mathop{\mathbb{E}}_{p_1,p_2,\tau} [W((s^*(p_1), s^*(p_2)); \tau) \mid p_1 \neq p_2] \right) \frac{d}{d\rho} \mathop{\Pr}_{p_1,p_2,\tau;\rho} [p_1 = p_2]
$$

$$
= \left( \mathop{\mathbb{E}}_{p_1,p_2,\tau} [W((s^*(p_1), s^*(p_2)); \tau) \mid p_1 = p_2] \right.
$$

$$
\left. - \mathop{\mathbb{E}}_{p_1,p_2,\tau} [W((s^*(p_1), s^*(p_2)); \tau) \mid p_1 \neq p_2] \right)
$$

$$
\cdot \frac{d}{d\rho} 1 - a_1 - a_2 + (1-\rho)a_1 a_2 + \rho \min(a_1, a_2)
$$

$$
\leq 0,
$$

where the last line follows by (1) and using the fact that

$$
\frac{d}{d\rho} 1 - a_1 - a_2 + (1-\rho)a_1 a_2 + \rho \min(a_1, a_2) \geq 0.
$$

This inequality is strict as long as $a_1, a_2 < 1$ (otherwise $\rho$ has no impact on the joint distribution of $p_1, p_2, \tau$). $\qquad \square$

## B.2 Proof of Theorem 4.2

*Proof.* The following condition for Firm 1 must hold for them to prefer prefer $\rho = \rho_B$ over $\rho = \rho_A$:

$$
\mathop{\mathbb{E}}_{p_1,p_2,\tau;\rho=\rho_B} [U_1((s^*(p_1), s^*(p_2)); \tau)] > \mathop{\mathbb{E}}_{p_1,p_2,\tau;\rho=\rho_A} [U_1((s^*(p_1), s^*(p_2)); \tau)].
$$

For ease of notation, let $A = \min(a_1, a_2) - a_1 a_2$ and let $\Delta_\rho = \rho_B - \rho_A > 0$. We will see that the probabilities cancel out when subtracting $\rho = \rho_B$ to $\rho = \rho_A$, leaving only the $A$ and $\Delta_\rho$ terms:

$$
\sum_{p_1 \in \{0,1\}} \sum_{p_2 \in \{0,1\}} \sum_{\tau \in \{\tau_H, \tau_L\}} U_1[(s^*(p_1), s^*(p_2)); \tau] [\mathbb{P}[p_1, p_2, \tau; \rho = \rho_B] - \mathbb{P}[p_1, p_2, \tau; \rho = \rho_A]] > 0
$$

$$
\frac{H\theta A \Delta_\rho}{2} - H\sigma\theta A \Delta_\rho + \frac{L(1-\theta)A\Delta_\rho}{2} - L(1-\theta)A\Delta_\rho - L\theta(1-\sigma)A\Delta_\rho + \frac{L\theta A\Delta_\rho}{2} > 0
$$

$$
\frac{1}{2} A\Delta_\rho [H\theta(1-2\sigma) + L(2\sigma\theta - 1)] > 0
$$

We can derive the same exact inequality for firm 2. When $\min(a_1, a_2) - a_1 a_2 \neq 0$ (or, when both $a_1, a_2 < 1$), we get

$$
\sigma < \frac{H\theta - L}{2\theta(H-L)}.
$$

We can further show that lower $\sigma$ monotonically increases preference for correlation:

$$
\frac{\partial}{\partial \sigma} A\Delta_\rho [H\theta(1-2\sigma) + L(2\sigma\theta - 1) = 2A\Delta_\rho\theta(L-H),
$$

which is always negative because $L < H$ by definition. $\qquad \square$

## B.3    Proof of Theorem 5.1

*Proof.* The main intuition behind this proof is that an algorithm with performance $a_i$ can simulate an algorithm with lower performance $a_c$. Recall that we define $s^*$ to be the optimal strategy of following the algorithm. Let $s^{\sim*}$ be the strategy of doing the opposite of the algorithm's recommendations. We define $s'$ to be the following strategy:

$$s'(a_i) = \begin{cases} s^*(a_i), & \text{w.p. } q \\ s^{\sim*}(a_i), & \text{w.p. } 1-q, \end{cases}$$

where $q = \frac{a_c + a_i - 1}{2a_i - 1}$. The strategy $s'(a_i)$ is equivalent in expectation to $s^*(a_c)$ in terms of firm utility. To see this, **we will prove that the conditional distribution $\mathbb{P}[\tau|s'(a_i)]$ is equivalent to $\mathbb{P}[\tau|s^*(a_c)]$:**

$$\mathbb{P}[\tau = \tau_H | s'(a_i) = 1] = \mathbb{P}[\tau = \tau_H | s^*(a_c) = 1]$$

$$\frac{\mathbb{P}[s'(a_i) = 1|\tau = \tau_H]\mathbb{P}[\tau_H]}{\mathbb{P}[s'(a_i) = 1|\tau = \tau_H]\mathbb{P}[\tau_H] + \mathbb{P}[s'(a_i) = 1|\tau = \tau_L]\mathbb{P}[\tau_L]} = \frac{\mathbb{P}[s^*(a_c) = 1|\tau = \tau_H]\mathbb{P}[\tau_H]}{\mathbb{P}[s^*(a_c) = 1|\tau = \tau_H]\mathbb{P}[\tau_H] + \mathbb{P}[s^*(a_c) = 1|\tau = \tau_L]\mathbb{P}[\tau_L]}$$

and

$$\mathbb{P}[\tau = \tau_H | s'(a_i) = 0] = \mathbb{P}[\tau = \tau_H | s^*(a_c) = 0]$$

$$\frac{\mathbb{P}[s'(a_i) = 0|\tau = \tau_H]\mathbb{P}[\tau_H]}{\mathbb{P}[s'(a_i) = 0|\tau = \tau_H]\mathbb{P}[\tau_H] + \mathbb{P}[s'(a_i) = 0|\tau = \tau_L]\mathbb{P}[\tau_L]} = \frac{\mathbb{P}[s^*(a_c) = 0|\tau = \tau_H]\mathbb{P}[\tau_H]}{\mathbb{P}[s^*(a_c) = 0|\tau = \tau_H]\mathbb{P}[\tau_H] + \mathbb{P}[s^*(a_c) = 0|\tau = \tau_L]\mathbb{P}[\tau_L]}.$$

Based on the Bayes' Rule expansion above, it suffices to prove the following equivalences:

$$\mathbb{P}[s'(a_i) = 1|\tau = \tau_H] = \mathbb{P}[s^*(a_c) = 1|\tau = \tau_H] \tag{4}$$

$$\mathbb{P}[s'(a_i) = 1|\tau = \tau_L] = \mathbb{P}[s^*(a_c) = 1|\tau = \tau_L] \tag{5}$$

Proof of (4):

$$\begin{aligned}
\mathbb{P}[s'(a_i) = 1|\tau = \tau_H] &= q\mathbb{P}[s^*(a_c) = 1|\tau = \tau_H] + (1-q)\mathbb{P}[s^{\sim*}(a_c) = 1|\tau = \tau_H] \\
&= q\mathbb{P}[s^*(a_c) = 1|\tau = \tau_H] + (1-q)\mathbb{P}[s^*(a_c) = 0|\tau = \tau_H] \\
&= qa_i + (1-q)(1-a_i) = \frac{a_c + a_i - 1}{2a_i - 1}a_i + \frac{a_i - a_c}{2a_i - 1}(1-a_i) = \frac{a_c(2a_i - 1)}{2a_i - 1} = a_c \\
&= \mathbb{P}[s^*(a_c) = 1|\tau = \tau_H]
\end{aligned}$$

Proof of (5):

$$\begin{aligned}
\mathbb{P}[s'(a_i) = 1|\tau = \tau_L] &= q\mathbb{P}[s^*(a_c) = 1|\tau = \tau_L] + (1-q)\mathbb{P}[s^{\sim*}(a_c) = 1|\tau = \tau_L] \\
&= q\mathbb{P}[s^*(a_c) = 1|\tau = \tau_L] + (1-q)\mathbb{P}[s^*(a_c) = 0|\tau = \tau_L] \\
&= (1-q)a_i + q(1-a_i) = \frac{a_i - a_c}{2a_i - 1}a_i + \frac{a_c + a_i - 1}{2a_i - 1}(1-a_i) = \frac{(2a_i - 1)(1-a_c)}{2a_i - 1} = 1 - a_c \\
&= \mathbb{P}[s^*(a_c) = 1|\tau = \tau_L]
\end{aligned}$$

Note that the space of possible accuracies is $a \geq 0.5$ for an algorithm to be useful. When $a_c = 0.5$, $a_i > 0.5$ by assumption of the Theorem and therefore $q$ is never undefined. Then,

$$E^1_{\rho_0}[(s^*(a_i), s^*(a_i))] \geq E^1_{\rho_0}[(s'(a_i), s^*(a_i))] = E^1_{\rho_0}[(s^*(a_c), s^*(a_i))],$$

and similarly for Firm 2.

$\square$

## B.4 Proof of Corollary 5.2

*Proof.* We will show that the condition for a strict preference for correlation (in the second-stage game) is equivalent to correlation being strictly in equilibrium (in the first-stage game). We first start with the preference correlation in the proof for Theorem 4.2:

$$\underset{p_1,p_2,\tau;\rho=\rho_B}{\mathbb{E}}[U_1((s^*(p_1),s^*(p_2));\tau)] > \underset{p_1,p_2,\tau;\rho=\rho_A}{\mathbb{E}}[U_1((s^*(p_1),s^*(p_2));\tau)].$$

Since this condition is for any $\rho_A < \rho_B$, we will let $\rho_B = \rho_c$ and $\rho_A = 0$. Further, we will change the $p$ notation to $a_c$ and $a_i$ where relevant, since $a_i = a_c$ by assumption.

$$\underset{a_c,a_c,\tau;\rho=\rho_c}{\mathbb{E}}[U_1((s^*(a_c),s^*(a_c));\tau)] > \underset{a_i,a_c,\tau;\rho=0}{\mathbb{E}}[U_1((s^*(a_i),s^*(a_c));\tau)],$$

which is equivalent to the condition that both firms using correlated models is in equilibrium; this strict inequality implies a strict equilibrium. Symmetric argument applies for Firm 2. $\qquad\square$

## B.5 Proof of Theorem 5.3

*Proof.* First, both firms using independent algorithms is always a PNE when $a_i > a_c$ as per Theorem 5.1.

We will next state what is needed to prove the theorem. When firms have a preference for correlation at $a_i = a_c$, both firms using correlated algorithms should be a PNE when $a_i = a_c + \epsilon$, for small enough $\epsilon$:

$$\exists\, \epsilon > 0 \text{ s.t. } E^1_{\rho_c,s^*}(a_c,a_c) > E^1_{\rho_0,s^*}(a_i+\epsilon,a_c) \tag{6}$$

On top of that, firms also prefer correlated algorithms over independence at $a_i = a_c + \epsilon$, for small enough $\epsilon$:

$$\exists\, \epsilon > 0 \text{ s.t. } E^1_{\rho_c,s^*}(a_c,a_c) > E^1_{\rho_0,s^*}(a_i+\epsilon,a_i+\epsilon). \tag{7}$$

The proofs for (6) and (7) come from Corollary 5.2, which states that when firms strictly prefer correlation at $a_i = a_c$, correlating is $\delta$-strictly a PNE:

$$\exists\, \delta > 0 \text{ s.t. } E^1_{\rho_c,s^*}(a_c,a_c) \geq E^1_{\rho_0,s^*}(a_i,a_c) + \delta,$$

We define the following shorthand:

| | |
|---|---|
| $A$ | $E^1_{\rho_c,s^*}(a_c,a_c)$ |
| $B$ | $E^1_{\rho_0,s^*}(a_i,a_c)$ |
| $C(\epsilon)$ | $E^1_{\rho_0,s^*}(a_i+\epsilon,a_c)$ |
| $D(\epsilon)$ | $E^1_{\rho_0,s^*}(a_i+\epsilon,a_i+\epsilon)$ |

Put another way, Corollary 5.2 states that

$$\exists\, \delta > 0 \text{ s.t. } A \geq B + \delta \tag{8a}$$
$$A \geq C(\epsilon) + \delta \tag{8b}$$
$$A \geq D(\epsilon) + \delta \tag{8c}$$

at $\epsilon = 0$ and $a_i = a_c$ because $B = C(\epsilon = 0) = D(\epsilon = 0)$. Since $C(\epsilon)$ and $D(\epsilon)$ are continuous in $\epsilon$, (6) and (7) are true by (8b) and (8c).

$\qquad\square$

## C Experiments (continued)

### C.1 Additional Details for Model Multiplicity Setup

We chose the following model hyperparameters to simulate a higher performance for random forests compared to logistic regression:

| Model | Hyperparameters |
|---|---|
| Logistic Regression | $\ell$1-penalty, saga solver |
| Random Forest | · # trees = 9 |
| | · min # samples in each leaf = 7 |
| | · weight: 1.2x for negative class |

All unspecified hyperparameters use the default values set by scikit-learn. All experiments (including the ones outlined in the following Section) were run using a Apple Silicon M2 chip with 16GB. They only require CPUs and are not computationally expensive – any modern computer can easily run these experiments.

## C.2 Additional Experiments: Data Procurement

### C.2.1 Setup

Our experiment involves two firms who may independently choose to correlate with each others' models by using overlapping datasets. Firm 1 trains on Census data in Texas while Firm 2 trains on Census data in Florida. They both have the option to purchase supplementary data **of worse quality** from a third-party, which in this case is Census data from California whose labels have been perturbed 25% of the time. In doing so, we are giving firms the choice of correlating their models at the expense of predictive accuracy.

In order to smoothly interpolate between independence and correlation, we define a parameter $\gamma$; for instance, Firm 1 can use the training data $(1 - \gamma)$ TX $+ \gamma$ CA, and similarly for Firm 2. If both firms use $\gamma = 0$, there is no overlap in training data and their resulting models will be the least correlated. Conversely, when both firms use $\gamma = 1$, their training data is identical and their models will be the most correlated.

We randomly sample $n = 200,000$ datapoints from TX, FL, and CA in order to standardize the effect of $\gamma$. We then further sample $\gamma$ proportion of each dataset to ensure that all training data used have exactly $n$ observations. We run this experiment over 15 random seeds, and over $\gamma \in [0, 1]$ in 0.1 increments. Both firms train the same model class (random forests) and have the same test data: Census data from Illinois.

### C.2.2 Results: Second-Stage Game

Figure 5(a) shows the predictive accuracy for both firms and the correlation between both firms as $\gamma$ varies. As expected, accuracy monotonically decreases and correlation monotonically increases as $\gamma$ increases since firms use more and more of the same lower-quality training data. We observe a significant decrease in accuracy for both firms when $\gamma = 1$, presumably because both models no longer receive the more predictive signal from their original training data.

Figure 5(b) shows the difference in utility between $\gamma$ at the x-axis and $\gamma = 0$ (independent datasets). When this difference is above 0 (blue dashed line), firms have a preference for correlation at that $\gamma$ value. We observe such a preference for correlation when consumers are more price sensitive (lower $\sigma$) and when the ratio between the $H$ and $L$ prices is larger, as per Theorem 4.2. Firms prefer correlation even when accuracy marginally decreases (subfigure (a)); this happens particularly when there is a high risk of being undercut, making correlating especially beneficial even at the expense of predictive accuracy. However, firms no longer prefer correlation when the trade-off in accuracy is too high (e.g., Firm 1 in $\gamma = 1$). We note that firms are asymmetric: because their models' accuracies differ at various $\gamma$, they do not always prefer correlation in the same way, but the general trends remain.

### C.2.3 Results: First-Stage Game

We also model firms' decision to correlate at a particular $\gamma$. In particular, Figure 6 shows the best response matrices for both firms in choosing various values of $\gamma$, over various model parameters $(H/L, \sigma)$. Cells with a red and blue cross indicate a Pure Nash Equilibrium. In general. higher correlation ($\gamma$) is only in equilibrium for higher $H/L$ and lower $\sigma$, which reflect the same trends as the second-stage game. For example, When $H/L = 6, \sigma = 0.1$, the sole equilibrium exists at

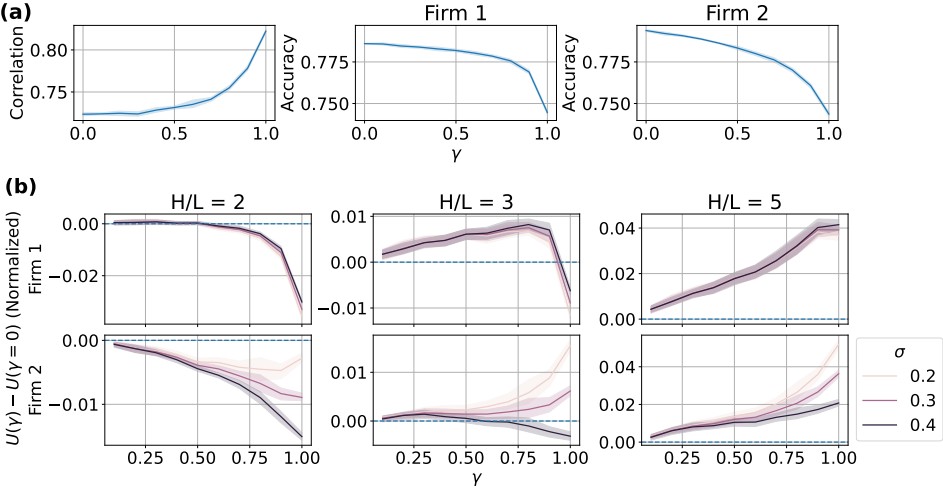

Figure 5: **(a)** [Left] Correlation between both firms' models in the empirical study across various values of $\gamma$. $\gamma = 0$ (1) corresponds to no overlap (full overlap) in training data. [Middle and Right] Accuracy of Firm 1 and 2's models over various values of $\gamma$. Error bars are 95% confidence intervals over 15 seeds. **(b)** Difference in utility between $\gamma$ at the x-axis and $\gamma = 0$ (no overlap in training data) for the empirical study, over various values of $H/L$ and $\sigma$. Top and bottom rows correspond to Firm 1 and 2's utilities, respectively. Shaded regions indicate 95% confidence intervals over 15 seeds.

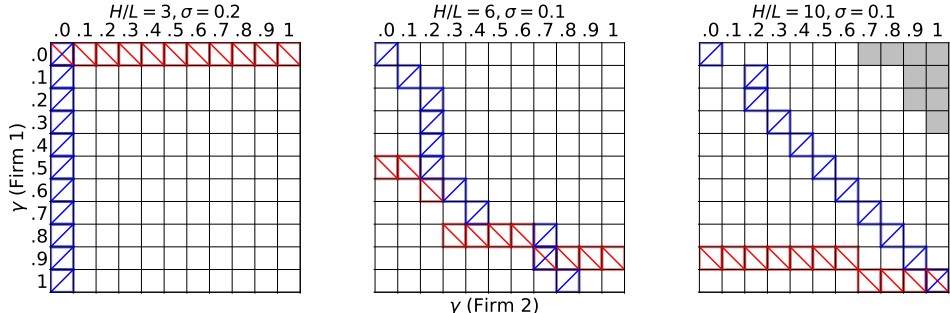

Figure 6: Best response matrices for the two firms in the empirical study, over three select model parameters. $\gamma = 0$ means no overlap in training data (least correlated) while $\gamma = 1$ indicates identical training data (most correlated). Best response for Firms 1 and 2 are highlighted in blue and red, respectively. Nash equilibria exist when both blue and red are highlighted in the same box (e.g., $(0,0)$ in the left subfigure). Grey boxes are "invalid" regions because following the algorithm would not have been a BNE in the downstream game where firms compete in prices. These results use the average firm utility over 10 seeds.

$(0.9, 0.7)$. When $H/L$ increases to 10, equilibrium is at $(1, 1)$. We note, however, that in extreme $H/L$ values, certain regions are "invalid" in the sense that firms would not follow the algorithm in the downstream second-stage game (grey cells).

## C.3 Additional Results

**Firms choose to correlate, even when algorithms are uninformative.** Figure 1 displays regions where firms following the algorithm's recommendation is a BNE for independent models only (light gray) and correlated models only (dark gray). When $a = 0.5$, independent models are never in equilibrium because the algorithms are as good as random. However, when $a = 0.5$ and models are correlated, firms may still choose to follow the algorithm. This region is more likely to be in low $\sigma$ regimes – where there is the highest risk in being undercut by one's opponent – and therefore there is value in coordinating actions despite the model having no predictive power.

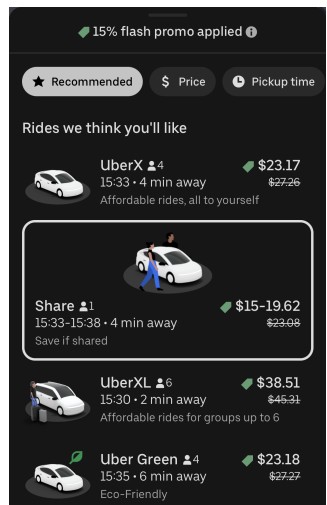 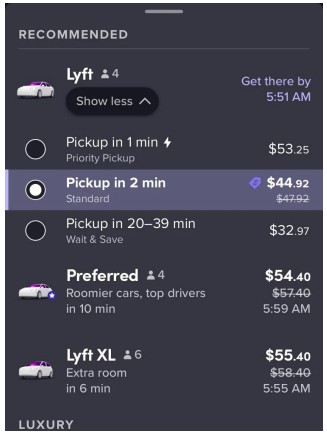 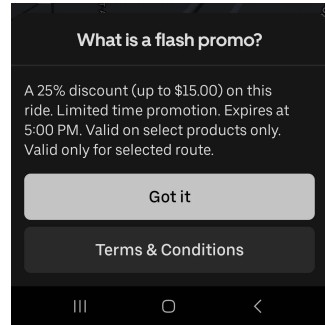

Figure 7: Examples of discounts offer to potential riders on Uber (left) and Lyft (middle). Rightmost figure explains Uber's "flash promo" offering.

# D   Extension: Asymmetric Market Power

In this section, we relax the assumption that consumers are indifferent to firms when firms offer the same price. In particular, we will introduce a parameter $\gamma \in [0, 1]$ that controls consumers' preference toward Firm 1. We will also reparameterize $\sigma \in [0, 1]$, still capturing price sensitivity. For example, in the context of Uber and Lyft, $\gamma$ specifies the proportion of people who check Uber before Lyft. Of the people who check Uber first, a $\sigma$ proportion are price-insensitive, meaning that they would still choose Uber even when Lyft offers a lower price. When $\gamma = 0.5$, we model the same consumer behavior as in the original model. The payoff matrices are summarized in Table 5.

|  | | Firm 2 | |
| --- | --- | --- | --- |
|  | | $H$ | $L$ |
| $[\tau_H]$ Firm 1   $H$ | | $(\gamma H, (1-\gamma)H)$ | $(\gamma\sigma H, (1-\gamma\sigma)L)$ |
| $L$ | | $((1-\sigma+\gamma\sigma)L, (1-\gamma)\sigma H)$ | $(\gamma L, (1-\gamma)L)$ |

|  | | Firm 2 | |
| --- | --- | --- | --- |
|  | | $H$ | $L$ |
| $[\tau_L]$ Firm 1   $H$ | | $(0,0)$ | $(0, L)$ |
| $L$ | | $(L, 0)$ | $(\gamma L, (1-\gamma)L)$ |

Table 5: Payoff matrices for both players when the consumer is willing to pay the high price ($\tau_H$, top) and low price ($\tau_L$, bottom). Within each cell, we denote (Firm 1 payoff, Firm 2 payoff).

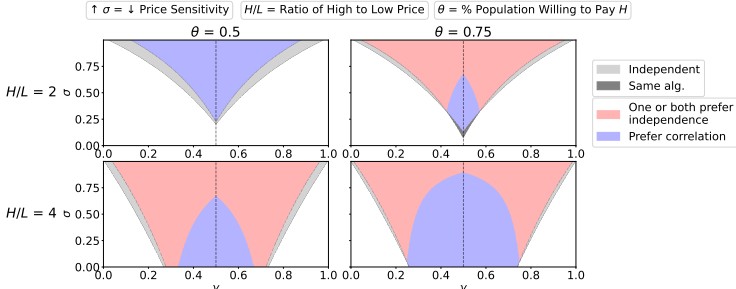

Figure 8: Regions where firms following the algorithm's recommendation is a Bayes Nash Equilibrium (BNE) for independent models only ($\rho = 0$, light gray), identical models only ($\rho = 1$, dark gray), and both (gradient). The blue region denotes where both firms have higher utility under $\rho = 1$, while the red region is where at least one firm has higher utility under $\rho = 0$. Columns represent two values of $\theta \in \{0.5, 0.75\}$, while rows represent two values of $H/L \in \{2, 4\}$. The x-axis in each subfigure is $\gamma$ and the y-axis is $\sigma$. We fix $a = a_1 = a_2 = 0.9$.

**Results.** Our main result is that **asymmetry lessens the impact of homogenization**. Figure 8 shows regions where using fully independent algorithms (light gray) and the same algorithm (dark gray) are BNE. When both are BNE, blue indicates that both firms have higher utility when using the same algorithm, and red otherwise. As $\gamma$ increases, firm 1 increasingly prefers correlation because they benefit from pricing similarly to their opponent (since consumers are increasingly loyal to firm 1). Firm 2 increasingly prefers independence as $\gamma$ increases for the same reason; firm 2 would like opportunities to undercut firm 1 since consumers prefer firm 1 when they price similarly. As a result, the region where both firms prefer correlation exist within a ball around $\gamma \in [0.5 - \epsilon, 0.5 + \epsilon]$ for some $\epsilon$.

# E   Extension: $n$ Firms

We now extend our model to capture a market with $n$ competing firms. Let $p_1, \ldots, p_n$ be the predictions of $n$ players. For simplicity, we assume that firms have two choices: to use a fully independent algorithm ($\rho = 0$) or to use the same (fully correlated) algorithm ($\rho = 1$). In Appendix E.4, we extend this to the setting in which firms can be partially correlated. We will also assume that all firms have the same model performance $a$.

Fix $n$ firms. Let $k \leq n$ be a "coalition" of firms who choose to employ the same algorithm, while the remaining $n - k$ firms are fully independent. Since all firms have the same accuracy, it does not matter which $k$ players we assign as part of the coalition. Hence, we will let $\Omega_k$ denote the joint distribution over $n$-bit vectors in which the first $k$ coordinates are identical, the last $n - k$ are mutually independent and independent from the first $k$. We explicitly define the joint distribution below:

$$\mathbb{P}(p_1, \ldots, p_k, p_{k+1}, \ldots, p_n | \tau_H) = a^z (1-a)^{1-z} \prod_{i=k+1}^{n} a^{p_i} (1-a)^{1-p_i}$$

$$\mathbb{P}(p_1, \ldots, p_k, p_{k+1}, \ldots, p_n | \tau_L) = a^{1-z} (1-a)^z \prod_{i=k+1}^{n} a^{1-p_i} (1-a)^{p_i}$$

where $a = \mathbb{P}(p_i = 1 | \tau_H) = \mathbb{P}(p_i = 0 | \tau_L)$ as in the main text, and $z = p_1 = p_2 = \cdots = p_k$ since the first $k$ firms predict the same outputs almost surely. In other words, whenever $p_1, \ldots, p_k$ are different from each other, the joint probability is 0.

## E.1   Firm Utility

Let $n_\ell$ and $n_h$ be the number of players that price low and high, respectively, such that $n = n_\ell + n_h$.

|  |  | All other firms but $i$ | | |  |  | All other firms but $i$ | |
|---|---|---|---|---|---|---|---|---|
|  |  | All same as $i$ | Some different |  |  |  | All same as $i$ | Some different |
| $[\tau_H]$ Firm $i$ | $H$ | $H/n$ | $\sigma H/n_h$ | | $[\tau_L]$ Firm $i$ | $H$ | $0$ | $0$ |
|  | $L$ | $L/n$ | $(1-\sigma)L/n_\ell$ | |  | $L$ | $L/n$ | $L/n_\ell$ |

Table 6: Payoff matrices for firm $i$ when the consumer is willing to pay the high price ($\tau_H$, left) and low price ($\tau_L$, right).

## E.2 First-stage and Second-stage game

We now define our equilibria of interest. For every $k$, we establish the following conditions:

**Second stage game (Deciding to use the algorithm).** The Bayes Nash Equilibrium condition is:

$$\mathbb{E}_{\mathbf{p}_{-i}\sim\Omega_k,\tau}[U_i((H,s^*(\mathbf{p}_{-i}));\tau) \mid p_i=1] \geq \mathbb{E}_{\mathbf{p}_{-i}\sim\Omega_k,\tau}[U_i((L,s^*(\mathbf{p}_{-i}));\tau) \mid p_i=1], \quad \forall i$$

$$\mathbb{E}_{\mathbf{p}_{-i}\sim\Omega_k,\tau}[U_i((L,s^*(\mathbf{p}_{-i}));\tau) \mid p_i=0] \geq \mathbb{E}_{\mathbf{p}_{-i}\sim\Omega_k,\tau}[U_i((H,s^*(\mathbf{p}_{-i}));\tau) \mid p_i=0], \quad \forall i,$$

where $\mathbf{p}_{-i}$ denotes the predictions of all $n$ firms save for firm $i$.

**First stage game.** We analyze whether or not the $k$ coalition is stable, given that in the downstream second-stage game firms follow the algorithm. Let $\mathcal{C}:=\{1,\ldots,k\}$ be the set of firms that use the same algorithm, and $\mathcal{I}:=\{k+1,\ldots,n\}$ be the set of firms that are fully independent. We define $V_I(k)$ to be the utility of a firm that uses an independent model, and $V_C(k)$ as the utility of a firm in the coalition using the same algorithm, i.e.,

$$V_I(k) = \mathbb{E}_{\mathbf{p}\sim\Omega_k,\tau}[U_i(s^*(\mathbf{p});\tau)], \quad i \in \mathcal{I}$$

$$V_C(k) = \mathbb{E}_{\mathbf{p}\sim\Omega_k,\tau}[U_i(s^*(\mathbf{p});\tau)], \quad i \in \mathcal{C}$$

The Nash equilibrium condition is:

$$V_I(k) \geq V_C(k+1) \tag{9a}$$
$$V_C(k) \geq V_I(k-1). \tag{9b}$$

In other words, players not in the coalition should have higher utility being independent and not join the coalition. Players in the coalition should be satisfied with staying in the coalition compared to their option of leaving.

*Special case when $k = 0$ and $k = n$.* Note that when $k = 0$, only condition (9a) applies. It is always an equilibrium, since if no one is using the fully correlated algorithm, no player can strictly increase their utility by switching from being fully independent to fully correlated. $k = 1$ can also be an equilibrium, but only if none of the $n-1$ independent players strictly benefit by choosing the correlated algorithm. When $k = n$, only condition (9b) applies.

## E.3 Results

Figure 9 shows regions where firms using their algorithm are in equilibrium for both the first and second stage games, for varying number of competing firms $n$ and model parameters. We note first that the regions $k > 1$ are disjoint. This is because the first-stage conditions (Equations (9b) and (9a)) are monotonic in $k$ and thus create disjoint regions.

**Larger coalitions ($k$) are only stable when consumers are price sensitive and are willing to pay the high price.** For example, in Figure 9, the red ($k = 4$) and purple ($k = 5$) regions exist only when $\sigma$ is low and $\theta$ is high. Intuitively, this makes sense because the larger the coalition, the more risk there is; deviating means potentially being able to undercut the coalition when they make a mistake. Large coalitions are therefore most valuable/preferable when consumers are incredibly price sensitive and most consumers are willing to pay the high price to begin with.

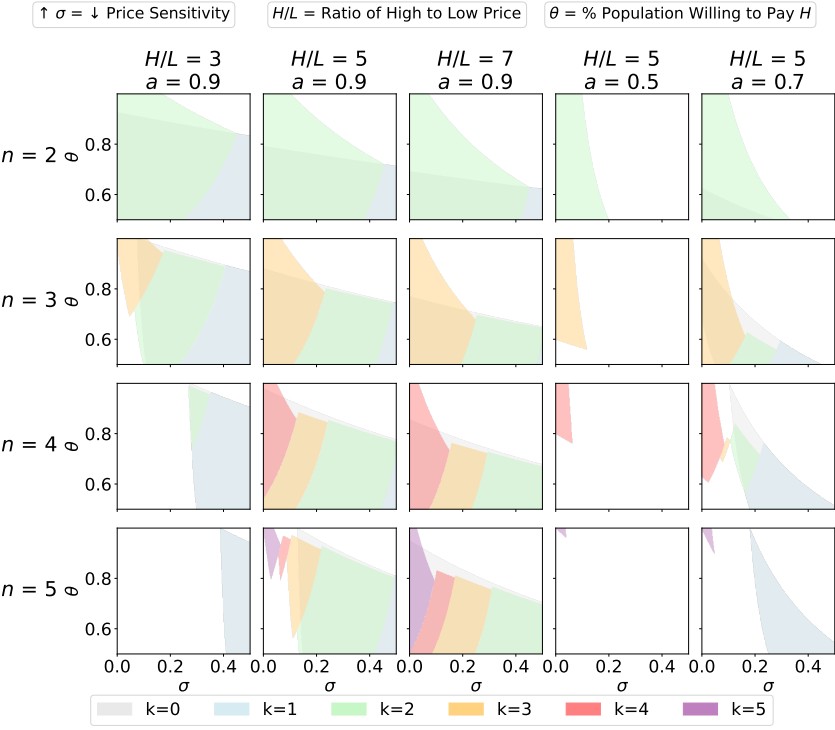

Figure 9: Regions where firms are in equilibrium for both the first and second-stage games, for various coalition sizes $k$. From top to bottom, each row represents an increasing number of competing firms $n$. Columns represent different model parameters $H/L$ and $a$.

**With many competing firms, correlation is only stable with high-performing models and high price differentiation.** For example, in Figure 9, the $n = 4, 5$ settings only have significant colored regions when $a = 0.9$ and $H/L = 5, 7$ (the highest options). The intuition is that at high $H/L$ (e.g., 7), there is lower risk of being undercut since a firm will still get 7 times the surplus when they make the sale. Also, when firms have a highly accurate model, they make fewer mistakes and hence have fewer chances of being undercut. Hence, they can correlate without fear of being undercut.

### E.4 Modeling Correlations in Full Generality: Gaussian Copula

In our analysis above, we restricted firm interactions to either be fully correlated or fully independent with each other. We also required that there exists only one coalition, and that firms employ algorithms with the same accuracy. Here, we provide a model that relaxes all three restrictions.

Assume that $n$ firms are divided into $m$ disjoint clusters $C_j$, $j \in [m]$. The main idea is that *(1)* correlations are constant within each cluster; and *(2)* we are only modeling pairwise correlations for all players and no higher order terms. In particular, we introduce correlation parameters $\rho_g$ and $\rho_{C_j}$, for all $j \in [m]$, which denote inter-cluster correlations and within-cluster correlations in $C_j$, respectively. Mapping back to our analysis above, we had $k$ of $n$ players form one cluster with $\rho_{C_1} = 1$ and the remaining $n - k$ players form singleton clusters where the within-cluster correlation is trivial and $\rho_g = 0$.

We will model each binary outcome $p_i$ using a latent Gaussian variable $Z_i$:

$$p_i \mid \tau_H = \begin{cases} 1, & Z_i \leq t_i \\ 0, & Z_i > t_i \end{cases} \qquad\qquad p_i \mid \tau_L = \begin{cases} 1, & Z_i > t_i \\ 0, & Z_i \leq t_i \end{cases}$$

where $t_i = \Phi^{-1}(a_i)$. This way, we still preserve the property that $\mathbb{P}(p_i = 1 \mid \tau_H) = \mathbb{P}(p_i = 0 \mid \tau_L) = a_i$. To model the joint distribution, we first define a multivariate normal distribution $Z \sim N(0, \Sigma)$, where $\Sigma$ is 1 in the diagonals, $\rho_{C_k}$ for pairs within cluster $C_k$, and $\rho_g$ for pairs in different clusters.

For example, if $n = 3$ and firms 1 and 2 are in a coalition, then the covariance matrix $\Sigma$ will be[8]:

$$\begin{bmatrix} 1 & \rho_{C_1} & \rho_g \\ \rho_{C_1} & 1 & \rho_g \\ \rho_g & \rho_g & 1 \end{bmatrix}$$

Let the probability distribution function (PDF) of $Z$ be $\phi_\Sigma$. Then, we let

$$\mathbb{P}(p_1, \ldots, p_n | \tau) = \int_{B_1} \cdots \int_{B_n} \phi_\Sigma(z_1, \ldots, z_n) dz_1 \ldots dz_n.$$

where

$$B_i = \begin{cases} (-\infty, t_i] & \text{if } p_i = 1 \\ [t_i, \infty) & \text{if } p_i = 0 \end{cases} \quad \text{if } \tau = \tau_H \qquad B_i = \begin{cases} (-\infty, t_i] & \text{if } p_i = 0 \\ [t_i, \infty) & \text{if } p_i = 1 \end{cases} \quad \text{if } \tau = \tau_L$$

This ensures that all combinations of $p_1, \ldots, p_n$ gives probabilities that sum up to one.

### E.4.1 Converting $\rho$ from binary space to Gaussian space

Ideally, we would like to specify $\rho_g$ and $\rho_{C_k}$ in binary space because it is our outcome of interest. However, as described above, the $\rho$ need to in the covariance matrix of the multivariate normal. This means we need a way to translate from $\rho_{\text{binary}}$ to $\rho_{\text{gaussian}}$. Since we are only modeling pairwise correlations, this mapping can be done for every pair of players. In particular, we will use the polychoric correlation approach [20], which establishes a relationship between the correlation of two ordinal variables, each assumed to represent latent bivariate Gaussian variables.

$$\rho_{\text{binary}} \simeq \frac{\int_{-\infty}^{t_i} \int_{-\infty}^{t_j} \phi_{\rho_{\text{gaussian}}}(z_i, z_j) dz_i dz_j - a_i a_j}{\sqrt{a_i(1 - a_i)a_j(1 - a_j)}},$$

where $\phi_{\rho_{\text{gaussian}}}$ is the joint bivariate normal PDF with correlation $\rho_{\text{gaussian}}$. We leave analyses involving these more complex interactions to future work.

---

[8]Note that by definition of the multivariate normal, the associated covariance matrix $\Sigma$ must be positive semi-definite, which is generally not satisfied if $\rho_C = 1$ and $\rho_g = 0$. We can set $\rho_C = 1 - \epsilon$ for some small $\epsilon > 0$ as an approximation for $\rho_C = 1$ in order to satisfy the positive semi-definite condition.

