# OpenReview forum: "Homogeneous Algorithms Can Reduce Competition in Personalized Pricing"
_NeurIPS.cc/2025/Conference — NeurIPS 2025 poster_

### Official Review · Reviewer_Z8sD · 2025-06-20

**Clarity:** 3
**Significance:** 2
**Originality:** 3
**Rating:** 4
**Confidence:** 3

**Summary:**

This paper investigates a mechanism for collusion between competing firms using personalized pricing algorithms. The authors propose that firms can reduce price competition by using "homogeneous" algorithms—that is, algorithms that produce correlated predictions. The paper develops a formal game-theoretic model of a duopoly where two firms use algorithms to segment consumers and decide whether to offer a high or a low price.

The main theoretical findings are: (1) higher correlation between firms' algorithms harms consumer welfare, and (2) firms are more incentivized to use correlated algorithms when consumers are highly price-sensitive. The authors show that firms may strategically prefer to use a less accurate but more correlated algorithm over a more performant but independent one. These theoretical results are demonstrated in an experiment using public census data.

**Questions:**

* How robust are the findings to a more realistic pricing model? If firms could choose prices from a continuous range, would the incentive to sacrifice accuracy for correlation still exist, or would competitive pressures dominate?
* The sigma parameter for consumer choice is essential to the results. How would the equilibria change under alternative models of consumer behavior, such as those involving search costs, network effects, or reference effects?
* The paper argues that its static model is a feature that isolates the correlation mechanism. However, couldn't this mechanism be amplified in a dynamic setting? For instance, could RL agents learn to choose simpler, more correlated models as a way to facilitate convergence to a collusive pricing policy?
* From a practical, regulatory standpoint, how can one distinguish between firms intentionally correlating via a less accurate model (the proposed "plus factor") versus firms that are simply less sophisticated and independently choose a simpler (e.g., linear) model that happens to be more correlated? The paper touches on this, but the problem of inferring intent seems very difficult.

**Ethical Concerns:**

["NO or VERY MINOR ethics concerns only"]

**Final Justification:**

As explained in the response to the reviewer, my overall assessment lies at the borderline. I think the paper has some merit, and also admit clear limitations. I am willing to follow the AC's final decision.

**Limitations:**

The authors correctly note some limitations. My main concerns are stated in the Weaknesses and Questions sections.

**Paper Formatting Concerns:**

* Table caption should appear before the table.
* Table needs to be centered. So I think Table 2 cannot be put like this.
* Table 1 is too wide, and its right side exceeds the text region.

**Quality:**

2

**Strengths And Weaknesses:**

### Strengths
* The paper studies an important problem in algorithmic fairness and economics—how shared components in ML pipelines can lead to anti-competitive outcomes. The focus on homogeneity as a specific mechanism for collusion is a good contribution.
* The work is well-motivated by current events and its connection to antitrust law is a strength. The discussion of legal implications is good.
* The paper effectively combines a formal game-theoretic model with an empirical study. The experiments, while simple, provide a demonstration that the theoretically-derived incentives can manifest with standard machine learning models on real-world data.

### Weaknesses
* The model's primary weakness is its simplicity. The assumption of a two-price world (High/Low) is a major abstraction from real-world pricing. The consumer choice model, governed by a single price-sensitivity parameter sigma, is also highly simplified. These abstractions make it difficult to be confident that the core results would hold in settings with continuous pricing or more complex consumer search behaviors.
* The paper relies on a static Bayes Nash Equilibrium analysis. This approach deliberately abstracts away from the learning dynamics through which firms might arrive at a collusive equilibrium. While this isolates the correlation mechanism, it also makes the model less descriptive of the "algorithmic" part of algorithmic collusion, which often implies adaptation and learning over time.
* The underlying principle (reduce strategic uncertainty through correlation can soften competition and lead to higher prices) is a known concept in the economics of oligopolies. The paper's novelty lies in applying this principle to the specific context of ML model development practices, which is a valuable but incremental step.

---

> ### Author Response · Authors · 2025-08-01
> **Rebuttal**
>
> Thank you for the helpful comments.
>
> Weakness 1 and Question 1. Since we are proposing a new mechanism of algorithmic collusion, we aim to demonstrate this phenomenon in a minimal working example. We did extend the base model by considering asymmetric market power (Appendix D) and n competing firms (Appendix E), which showed that the main findings generally hold with more complexity. In general, we expect various other extensions to still provide the same intuition as well.
>
> We would like to clarify that our problem is that of personalized discounting rather than setting base prices (which is a continuous problem and depends on first principles of supply and demand). In the context of discounting in personalized pricing, firms in reality typically consider discrete price options rather than continuous prices (see, e.g., Dubé & Misra, 2023 or Uber and Lyft discounts). If we were to model the continuous pricing regime, we would expect to see a race to the bottom where firms constantly undercut each other; hence we would need to make additional subjective modeling choices that we avoid in the first place with the discrete model.
>
> Weakness 2. We would like to clarify what our solution concept (BNE) captures. BNE (and equilibria in general) characterize the set of limiting behaviors in repeated games given all players are rational. The existence of a BNE solution (say, s^*) does not necessarily mean that firms will play that strategy in the one-shot setting, but rather that after sufficient rounds of interacting, s^* is one possible solution that firms will “settle” on and will not be incentivized to deviate. Therefore, BNE bakes in dynamic interactions through time. Collusive equilibria in the one-shot game we study are in a sense stronger than collusion in repeated games. Consider, for example, the (repeated) prisoner’s dilemma game. In the one-shot setting, the only equilibrium is non-collusive. But collusion is possible in repeated games because agents fear the threat of future retaliation. Indeed folk theorems in game theory show that the set of equilibria in repeated games can be much broader than those in one-shot games. Roughly speaking, collusive equilibria in the one-shot setting imply collusive equilibria in repeated games, but the converse isn’t true. We took great care to elucidate this point in lines 194-200 as this is what sets us apart from existing works on algorithmic collusion.
>
> Weakness 3. We agree with the reviewer that our basic conclusion – algorithmic collusion is bad for consumers – is relatively understood in economics. However, our paper is novel in that it reveals a novel mechanism for collusion: that firms can tacitly coordinate through algorithmic homogenization. Investigating specific mechanisms is particularly important when analyzing the legal implications of algorithms, since the law is concerned with both process and outcomes. Our model allows us to apply concepts from antitrust law to this new mechanism (Section 7.1), where we show for the first time that collusion due to algorithmic homogenization lies in a legal gray area despite potential for consumer harm. Therefore, we believe our insights are quite important and not obvious from previous literature.
>
> Question 2. In modeling $\sigma$, we are implicitly assuming that firms do not know which consumers shop around for a lower price. Under this assumption, many natural behavioral models reduce to ours. For example, if consumers have heterogeneous search costs (unknown to the firms), then only a fraction of them will be willing to shop around, yielding our model. We do not consider network structures on the consumers, though this could be an interesting future extension.
>
> Question 3. Our results are agnostic to precisely how the firms approach equilibria. In Bayesian games, no-regret learning dynamics can converge to equilibria (Hartline et al., 2015), so one can think of our model as describing the limiting behavior of learning agents interacting with one another. Our goal is to characterize these limiting behaviors as opposed to analyzing the dynamics of particular learning strategies, but we believe that our equilibrium concept captures the outcomes that reasonable learning dynamics would converge to. See also our response to Weakness 2 for a related discussion.
>
> Question 4. Great question! Our conclusion implies that proving intent in this situation is indeed very difficult, partly because there exists almost no legal precedent for these types of cases, and partly because antitrust law captures many complex scenarios with varying burdens of proof. While we discuss certain ways the law or regulation could evolve in our paper, we view our contribution as highlighting this novel mechanism of collusion in service of future work (both legal scholars and the courts/government agencies) to continue this important line of work.

---

> ### Comment · Reviewer_Z8sD · 2025-08-06
>
> Thank you for the rebuttal. My questions are partly solved. I understand that many pricing problems need to be studied in a simplified setting, otherwise it is not easy to derive some theoretical insights. However, I also agree with Reviewer zrwy's concern that the model studied is indeed too restricted to be applied in any practical senses.
>
> My overall assessment for the paper is that it lies just on the borderline of acceptance (or rejection), like between 3 and 4. Given the current rating distribution, I'd like to increase my rating to 4, otherwise I think there is no chance for the paper to be considered for acceptance.

---

### Official Review · Reviewer_zrwy · 2025-06-25

**Clarity:** 3
**Significance:** 2
**Originality:** 2
**Rating:** 3
**Confidence:** 4

**Summary:**

This paper studies the possibility of algorithmic collusion through using highly correlated predicting algorithms for price discrimination. The authors consider a duopoly model and model their competition with a few parameters including consumer’s price sensitivity ($\sigma$), extent of algorithm correlation $\rho$, price discrimination ratio $H/L$, the competitors’ prediction accuracy ($a_1$) and his own prediction accuracy $a_2$. Under this stylized model with two-firm, binary action, symmetric prediction error, the paper shows the following main results:
(1) Given these known parameters, when Bayesian Nash equilibrium exists, the authors prove that consumer welfare decreases when $\rho$ (correlation between two firm’s action) increases.
(2) When consumers are more price sensitive, firms prefer highly correlated algorithms than independent ones.
(3) An extended study about the situation where the firm can strategically choose an independent algorithm (by training on their own data) or a correlated algorithm (by buying data from vendors), and identifies conditions under which correlated algorithm is always preferred.
(4) They verify their sights through simulation on ACSIncome data.

**Questions:**

See weakness for concrete questions.

Some additional questions
-	What does the white region in Figure 1 mean?
-	In Table 1 caption, should “top” “bottom” be left, right?

**Ethical Concerns:**

["NO or VERY MINOR ethics concerns only"]

**Limitations:**

Yes.

**Paper Formatting Concerns:**

No.

**Quality:**

2

**Strengths And Weaknesses:**

Strength:
-	The problem studied in this paper is very interesting. Algorithmic collusion is indeed a critical, intriguing and difficult problem that touches AI, laws, policy and regulation
-	 The paper illustrate interesting insights into the issue of price discrimination from a tractable economic model. It highlights how algorithmic correlations, a growing concern in the AI domain, can negatively impact consumer welfare.
-	This paper leverages publicly available data and machine learning algorithms in a very smart way to efficiently demonstrate the insights behind its proposed method.

Weakness:
The reviewer has a very mixed feeling. While the reviewer understands that it is difficult to strike a balance between insights and model complexity, the reviewer feels the studied model is a bit too simplistic and several assumptions are too strong – a learning algorithm is summarized into a single number $a$, pricing actions is simplified into two numbers, and correlation is captured by a single number $rho$ (PS: I seem to be missing how explicitly $rho$ affects the correlation of $p_1, p_2$), and all the knowledge is publicly shared. While these simplifications make the models very tractable (I think most quantities can be characterized in closed form under this simple model), I am not sure how generalizable the obtained insights are.

On the other hand, the reviewer also found the conclusions derived from the model is mostly expected, which reduces the novelty and insightfulness of the paper.

To concretize my concerns above, the reviewer has the following questions:

-	True positive rate most likely is very different from true negative rate, especially when the class label (customer type) is unbalanced. Will the insights still hold if this assumption has been relaxed?
-	The definition of correlation is not rigorous. What is the mathematical formula of the correlation studied in this paper? Does this author consider the Pearson correlation, which is a ratio between condition covariance over conditional variance? The unrigorous definition leads to unrigorous proof of Theorem 4.1, which treats correlation as an exogenous variable, and independent of a1, a2, the performance of algorithms employed by the companies. Usually, a1, a2, and $rho$ moves at the same time, which the reviewer suggest to add simulations to study this problem

---

> ### Author Response · Authors · 2025-08-01
> **Rebuttal**
>
> Thank you for the questions!
>
> When true positive rate and true negative rate are not the same (often due to class imbalance as the reviewer points out), we might expect that the metric corresponding to the dominant class to be the relevant metric precisely because it applies to more consumers. Generally, separating TPR and TNR is a valid extension and can be modeled by adding one extra parameter; our framework can easily capture that.
>
> Regarding the question about the definition of correlation – throughout the paper we made sure to clarify that the quantity of interest is excess correlation, i.e., correlation that is unexplained by model performance. Rho is formally defined in Table 3; indeed, it does not correspond to the typical notions of correlation (e.g., Pearson) precisely because we are measuring prediction similarity in excess of performance. We note that in our case of two Bernoulli’s, all models of correlation are equivalent up to reparametrization.
>
> The reviewer also mentioned that we treat a1, a2, and rho as exogenous in Theorem 4.1, suggesting to treat them as endogenous instead. This is precisely what we model in Section 5, and we find similar results.
>
> In general, our model is indeed quite simple; since we are proposing a new mechanism, we aim to demonstrate this phenomenon in a minimal working example. We did extend the base model by considering asymmetric market power (Appendix D) and n competing firms (Appendix E). However, we did our best to justify any assumptions we made – for example, we argued that assuming shared knowledge of relevant parameters is standard and somewhat expected in oligopolies with frequent interactions, as is the case with e-commerce markets. Note also that our experiments relax many of these assumptions, and our theoretical findings still hold.
>
> Finally, the white regions in Figure 1 signify parameters where firms would not have used the algorithm in the first place (i.e., always price high or low, since the cost of mistakes does not matter when H/L is too high or too low). The caption in Table 1 is a typo and we will fix it – thank you for pointing it out!

---

> > ### Comment · Reviewer_zrwy · 2025-08-04
> > **Thanks for the Rebuttal**
> >
> > Appreciate the authors' rebuttal which does clarify the parts that I was confused.
> >
> > A somewhat tough choice for me to make, but after some debate with myself, I think I will keep my initial rating  since I think my major concern about the model form being over-simplistic (with noting the attempt of generalizations in Appendix D,E) and the conclusion being quite expected still remains, and I am not sure whether there is an easy fix to this.

---

> > > ### Author Response · Authors · 2025-08-05
> > > **Clarification on Main Contributions and Novelty**
> > >
> > > Thank you for the continued discussion! We would like to clarify that even if the reviewer finds the technical conclusions as expected, **our primary goal is to reveal a novel mechanism for collusion through homogenization**. Investigating specific mechanisms is particularly important when analyzing the legal implications of algorithms, since **the law is concerned with both process and outcomes**. Our model allows us to apply concepts from antitrust law to this new mechanism (Section 7.1), where we show for the first time that collusion due to algorithmic homogenization lies in a legal gray area despite potential for consumer harm. We would also add that the empirical section validates our theoretical findings while relaxing many of the simplifying assumptions we made in our model.
> > >
> > > In sum, we view our simple model as a vehicle to introduce a novel mechanism of collusion, which has important legal implications.

---

### Official Review · Reviewer_jsLv · 2025-07-02

**Clarity:** 3
**Significance:** 3
**Originality:** 3
**Rating:** 4
**Confidence:** 3

**Summary:**

This paper studies the problem of algorithmic homogeneity in personalized pricing. Specifically, it proposes a game-theoretic model to demonstrate how correlated algorithms can facilitate tacit collusion among competing firms, ultimately harming consumer welfare. Empirical studies on a stylized game between two firms are presented to demonstrate the theoretical results. Finally, the paper connects its technical findings to current antitrust law, suggesting new plus factors for detecting collusion.

**Questions:**

The model simplifies pricing to a binary choice. How would the results change if firms could set prices on a continuous spectrum, or if there were more than two discrete price points? Would the incentives for correlation still hold?

**Ethical Concerns:**

["NO or VERY MINOR ethics concerns only"]

**Final Justification:**

The authors satisfactorily addressed all of my questions. My rating remains unchanged.

**Limitations:**

yes

**Quality:**

3

**Strengths And Weaknesses:**

On the positive side, this paper presents a timely and important investigation into the anti-competitive implications of algorithmic homogeneity in personalized pricing. The game-theoretic model is well-defined, and the theorems clearly present the core findings: consumer welfare decreases with higher correlation, and firms are incentivized to choose correlated algorithms, especially when consumers are price-sensitive. I found the problem presented by the paper interesting and relevant. In general, this paper proposed an innovative research topic.

My main concern with this paper is its focus on the oversimplified model with two firms selling identical goods with a binary pricing scheme. While the extensions of asymmetric market power and N-firms are discussed in the appendix, it suggests that they are secondary. Given the importance of real-world market complexities, a more integrated discussion of how firm asymmetry and multiple firms impact the core findings (e.g., Theorems 4.1, 4.2, 5.3) could improve the paper's scope.

---

> ### Author Response · Authors · 2025-08-01
> **Rebuttal**
>
> Thank you to the reviewer for the helpful comments. To address your concern, we added the extensions of asymmetric market power and $N$ firms as a robustness check that our main results and mechanisms hold even when we relax certain conditions. Our main text focuses on the simple case to shed light on a novel mechanism of collusion, which allows us to analyze its legal implications.
>
> To answer your question – if there were more than two discounting levels, we expect our main results to hold simply because correlation will always be a mechanism for tacit collusion. In the context of personalized discounting, firms in reality typically consider discrete price options rather than continuous ones (see, e.g., Dubé & Misra, 2023). If we were to model the continuous pricing regime, we would expect to see a race to the bottom where firms constantly undercut each other; hence we would need to make additional subjective modeling choices that we avoid in the first place with the discrete model. We would also need to rule out perfect price competition. While we currently do this with $\sigma$, we would need a much stronger set of assumptions for continuous-price models.

---

> > ### Comment · Reviewer_jsLv · 2025-08-06
> >
> > Thank you for your responses. I will take them into account during the AC-Reviewer discussion phase. I don’t have any further questions.

---

### Official Review · Reviewer_84Rt · 2025-07-02

**Clarity:** 4
**Significance:** 4
**Originality:** 4
**Rating:** 5
**Confidence:** 4

**Summary:**

* This paper develops a game-theoretic model of price discrimination and collusion through the choice of predictive models.
* In the setting, there are two firms selling identical goods. The goods may either sell at a high or low price leading to per-unit profits $H$ or $L$, respectively. A fraction $\\theta$ of consumers have valuations higher than the high price, and their sensitivity to price is characterized by a parameter $\\sigma$. Firms predict the type of each consumer (high/low valuation) using a classifier $p_i(x)$, with group accuracy characterized by a parameter $a_i$. Predications have correlation parameterized by $\\rho$. The strategy space of each firm is the price they offer given a prediction, and the goal is to characterize the equilibria of the resulting game.
* The main results show that correlation hurts user welfare (Theorem 4.1), and characterize the firm’s preference towards correlation as a function of price sensitivity (Theorem 4.2). The second set of results analyzes the resulting equilibria when firms can decide whether to train prediction models independently or using common data - Showing the firms may benefit from correlated but less accurate prediction models (Theorem 5.3). Empirical analysis is presented based on an income-level prediction task, showing that similar phenomena are also evident in real-world datasets.
* Finally, broader implications are discussed with an emphasis on legal implications.

**Questions:**

* In cases where user welfare is negatively affected by the strategic decisions of the firms, is it possible to incentivize the firms to act in a way which benefits the users?
* How does the work relate to the existing literature on AI and competition? (e.g., Ben-Porat & Tennenholtz, EC 2019; Jagadeesan et al., NeurIPS 2023; Einav & Rosenfeld, ICML 2025; Tsoy & Konstantinov, NeurIPS 2023; Laufer et al., WWW 2024, etc.). Does the interaction model generalize existing models in this literature? Is it possible to interpret some of these results from the perspective of collusion?
* What happens if other types of agents are introduced to the dynamics? For example, what are the implications of LLM providers being allowed to adjust their price dynamically based on demand?

**Ethical Concerns:**

["NO or VERY MINOR ethics concerns only"]

**Final Justification:**

This is a very interesting paper, and I especially appreciate its clear aim of informing public policy. The reviews have raised concerns about the model possibly being too simplistic, and helpful clarifications were provided in the rebuttal. Given this, I maintain my original score, and I believe that the paper can benefit from a more prominent discussion of the model’s scope, core assumptions, practical applicability, and limitations.

**Limitations:**

Limitations and broader impact are discussed.

**Paper Formatting Concerns:**

No formatting concerns.

**Quality:**

4

**Strengths And Weaknesses:**

Strengths:
* The paper is well-written and very easy to follow. Formal statements of results are clear, and intuition for the results is provided.
* The proposed model is intuitive and seems to capture key aspects of collusion dynamics in a concise way.
* Results offer actionable insights that have potential to influence policy decisions.

Weaknesses:
* Results seem to rely on the assumption that all the relevant information about the setting is known to all parties, while in practice such information may be hard to obtain. In particular it seems that estimation of the correlation factor $\\rho$ in practical settings would require sharing fine-grained model performance metrics between competitors. It’s unclear whether results hold in settings in which the core assumptions are not met.

---

> ### Author Rebuttal · Authors · 2025-07-31
>
> Thank you to the reviewer for the comments! We do indeed assume that firms have information about competing models’ accuracies and correlations. This assumption is realistic in oligopolies that frequently interact with each other (i.e., small but frequent purchases from many consumers, as is the case with our motivating scenarios of Uber and Lyft). Firms that frequently observe the market outcomes from their predictions allow them to better estimate the relevant parameters in equilibrium. Another perspective is that when competing firms play a no-regret strategy in this simple game, they will eventually converge to an equilibrium without explicit knowledge of parameters, see Hartline et al. (2015).
>
> Question (1). Interesting question. We suspect that the regulator would typically need enough information about firm behavior and the market in order to offer good incentives. However, regulators typically do not have this much information unless they conduct a formal investigation on possible antitrust violations.
>
> Question (2). In general, the cited references investigate different mechanisms/scenarios related to competition. For example, Einav & Rosenfeld (2025) show that instead of maximizing accuracy, firms might try to differentiate and segment the market – this outcome is of course possible, but many goods/services (e.g., Uber/Lyft, Amazon goods) are largely similar save for brand loyalty (which we can model via market asymmetry). We see a similar market segmentation effect in Jagadeesan et al. (2023). There are many scenarios that competitive firms face in real life, and we are merely analyzing a novel mechanism – one of many – that has important antitrust implications.
>
> Question (3). We believe our model can capture agents with different dynamics as well. The reviewer’s question about LLM providers adjusting prices would fit in our model; instead of making predictions about a user, they could be making decisions based on time, demand, etc. But of course, our model is limited to two possible prices (which can be easily extended to multiple and the same results will hold).

---

> ### Comment · Reviewer_84Rt · 2025-08-06
>
> Thank you for the detailed response! After reading the other reviews, I concur with the other reviewers' concerns about the model possibly being too simplistic. However, I also appreciate the paper's focus on legal implications, and acknowledge the benefit of conciseness in such settings.
>
> To help reconcile these perspectives, I wonder if the authors can further elaborate on the limitations of their simplifying assumptions. In particular:
> * Is it possible to describe concrete scenarios in which the proposed collusion mechanism is expected to occur, or is already believed to occur in some sense? If so, which model parameters (i.e., values of $p, \\rho, L, H$, etc.) best capture these scenarios? Is it possible to use the model to quantify the negative societal effects of collusion in these settings?
> * Conversely, would it be possible to describe scenarios in which the proposed collusion mechanism is theoretically possible, but is in practice likely to be dominated by other economic effects, or counteracted by other forces? (for instance, which additional economic effects could limit the applicability of the proposed collusion model in the "Uber-Lyft" scenario?)
>
> In addition, I also have a couple of additional (very minor) formatting remarks:
> * In Figure 3, the graph legends seem to hide some of the lines and error bars. Given the available whitespace around the plots, maybe it would be beneficial to relocate the legend for improved readability.
> * Some tables and equations appear to overflow outside the margins (e.g., Table 1, L510, L517, Table 6).

---

> > ### Author Response · Authors · 2025-08-07
> >
> > We appreciate the opportunity to further clarify our model’s usefulness and impact!
> >
> > **Point 1**. Yes! We suspect homogenization as a form of coordination is quite common in e-commerce. Apart from the Uber/Lyft model selection scenario from our paper, another scenario includes travel websites that use device information to price discriminate (e.g., Expedia and Booking have been known to charge Apple users more than Android users). We can view this as an instance of our model, with $\rho \approx 1$, since platforms have identical pricing strategies. $\sigma$ would matter less here since prices never differ.
> >
> > Another example is when multiple platforms use the same third-party data-sharing service (e.g., LiveRamp). In this case, firms access mostly the same information. $\rho$ could vary depending on model training decisions and how much private information they bring. We seek to capture this scenario in the data sharing experiments in Appendix C.2. Accuracy depends on the specifics of the application, but in our experiments, we find trade-offs between accuracy and correlation. More expressive model classes have higher variance, leading to less correlation and higher accuracy. Through A/B testing, a firm may discover that a more accurate model performs worse, since it is less correlated with the competitor.
> >
> > Regarding other parameters in our model—in the e-commerce settings we care about, consumers are often price sensitive (low $\sigma$) because they face little friction in switching between websites/apps. When there is a price difference, many consumers will shop around and pick the lowest price (especially since goods/services are largely undifferentiated). This is important since low $\sigma$ is what drives the incentive to correlate in Theorem 4.2. We also note that $H,L$ are per-unit profits and not posted prices; for example, $H/L=4$ means firms take a 75% hit in their profits should a sale occur, rather than consumers *seeing* a 75% discount on the posted price. We point this out because collusion is possible typically for higher $H/L$, which is still realistic since $H,L$ are not posted prices.
> >
> > Regarding your question about measuring societal effects—in theory, one can quantify welfare using Table 4. In practice, a third-party auditor will have to estimate these parameters by obtaining data about firms’ market interactions and consumer behavior; this may add complexity in future antitrust cases (e.g., many economics papers seek to estimate market power parameters for antitrust cases, see [1, 2]). But in simpler cases like the Expedia example of charging Apple users more, these parameters can be estimated easily (e.g., $\rho = 1$).
> >
> > **Point 2**. Great question! One possible effect our model does not capture is that firms might start to differentiate their goods/services, which segments the market naturally. Our results rely on the fact that these goods/services are interchangeable. However, in e-commerce especially, differentiation occurs mainly on the product side while we are more concerned about the platform side, which remains largely undifferentiated. For example, we see vastly different restaurant options available in both Uber Eats and Doordash (which are roughly interchangeable platforms). Even in the case where platforms do start to differentiate (e.g., Lyft advertises that their drivers get better wages than Uber), this mainly affects consumers’ loyalty to firms, which we capture in Appendix D. Note, however, that we are mainly referring to situations where platforms give discounts—which happens most often—but there are cases where sellers give discounts as well (e.g., Amazon), which is a secondary effect we do not capture.
> >
> > Another pricing strategy our model does not capture is that of membership fees (e.g., annual fees for Amazon Prime or Uber Eats). These membership options change consumer behavior because consumers face more complicated frictions between competing firms–say, if they have paid for Amazon Prime but not for Walmart+. In principle, incorporating membership fees in our model is possible, but it will likely no longer be a story about personalized discounting. On that note, we clarify that our work does not claim coordination in the targeted discounting problem is the only or dominant effect among competing firms. Other effects (like membership fees) might attenuate our findings; on the other hand, coordination through algorithmic homogenization likely exists in other problems outside of personalized pricing. Our primary goal is to isolate and analyze a novel effect–albeit one of potentially many–that has interesting legal implications.
> >
> > Thank you for pointing out the formatting issues; we will fix them before the camera-ready should the paper be accepted!
> >
> > [1] Alice L. Xu et al. “Market Power Abuse in Wholesale Electricity Markets.” (2025), arXiv:2506.03808.
> >
> > [2] Feng Zhu and Marco Iansiti. “Entry into Platform‑Based Markets.” Strategic Management Journal 33, no. 1 (2012).

---

> > > ### Comment · Reviewer_84Rt · 2025-08-08
> > >
> > > Thank you for the helpful clarifications! I have no further questions at this point.

---

### Note · Authors · 2025-08-12

Thank you to all the reviewers for their comments and questions. To summarize, our paper’s main contribution is introducing and analyzing **a novel mechanism of algorithmic collusion**: homogenization as a way to tacitly coordinate among competing firms. To demonstrate this, we devise a simple model that we believe faithfully captures various e-commerce settings — we want to emphasize that extensions/relaxations to our model (as in our Appendix and our empirical results) will merely complicate the theory but **generally the main mechanism will still hold**. Our core results are intuitive and some aspects (e.g., coordination reduces consumer welfare) are well-known in Economics. However, **our work’s novelty is in its connection to algorithmic homogenization and its implications to antitrust law**; without a formal theoretical model, we cannot analyze how firms might reach collusive equilibria, which is important because **the law is often concerned with both process and outcomes**. As we argue, this analysis is quite timely because many competing algorithms today are homogeneous to varying degrees.

---

### Decision · Program_Chairs · 2025-09-17

**Decision:**

Accept (poster)

**Comment:**

This paper introduces a novel mechanism of tacit algorithmic collusion in personalized pricing. It shows that correlated algorithms reduce consumer welfare and that firms may strategically prefer less accurate but more correlated models. The work is further supported by a stylized empirical study and connected to antitrust law.

All reviewers appreciated the clarity of the model, the formal results, and the strong policy relevance. However, some noted weaknesses in the simplicity of the assumptions, the limited generalizability, and the fact that some conclusions echo known results in economics. The rebuttal convincingly clarified definitions (e.g., correlation), addressed robustness (extensions to asymmetric and multi-firm settings), and provided concrete real-world analogies (Uber/Lyft, Expedia/Booking).

Overall, I find the conclusions worth publishing and recommend acceptance.